# Unexpected limitation of tropical cyclone genesis by subsurface tropical central-north Pacific during El Niño

Cong Gao [1], Lei Zhou [1,2] ✉, Chunzai Wang [3], I.-I. Lin [4] & Raghu Murtugudde[5,6]

The vast tropical Pacific is home to the majority of tropical cyclones (TCs) which threaten the rim countries every year. The TC genesis is nourished by warm sea surface temperatures (SSTs). During El Niño, the western Pacific warm pool extends eastward. However, the number of TCs does not increase significantly with the expanding warm pool and it remains comparable between El Niño and La Niña. Here, we show that the subsurface heat content change counteracts the favorable SSTs in the tropical central-north Pacific. Due to the anomalous positive wind stress curl, the 26 °C isotherm shoals during El Niño over this region and the heat content diminishes in the tropical central-north Pacific, even though warm SST anomalies prevail. This negative correlation between SST and 26 °C isotherm depth anomalies is opposite to the positive correlation in the tropical eastern and western Pacific. This is critical because quantifying the dynamics of the subsurface ocean provides insight into TC genesis. The trend in TC genesis continues to be debated. Future projections must account for the net effect of the surface-subsurface dynamics on TCs, especially given the expected El Niño-like pattern over the tropical Pacific under global warming.

The genesis of tropical cyclones (TCs) requires energy from the warm ocean[1] as one of the necessary conditions, although the full dynamics controlling the TC frequency remain a persistent mystery[2]. Due to global warming, sea surface temperatures (SSTs) over the tropical Pacific have warmed significantly[3] even though the fate of the east–west gradient in the deep tropics is debated[4,5]. It has been proposed that the easterly trade winds tend to weaken due to the reduction of the zonal SST gradient[6], favoring an El Niño-like pattern in the tropical Pacific[7–10]. However, much debate has occurred over the opposite secular trend during the first decade of the 21st century where the east–west SST gradient and the trade winds have strengthened leading to what is referred to as the "hiatus"[11]. The "hiatus" was declared to have ended since approximately 2011, resulting in an El Niño-like pattern of warming[12]. Therefore, an examination on the contrasts in TC genesis between El Niño and La Niña, the two phases of El Niño-Southern Oscillation (ENSO), is an analog to advance our understandings of TC genesis under global warming.

As a dominant mode of natural climate variability, ENSO influences weather and climate globally[13]. Warm (cold) SST anomalies occur in the tropical eastern and central Pacific during El Niño (La Niña), due to a coupling between SSTs, surface winds, and the thermocline[14–17]. ENSO can modulate the number of TCs[18,19] by modulating the ocean and atmosphere states. The western Pacific warm pool expands significantly eastward during an El Niño[20]. However, the total number of TCs generated in the tropical western North Pacific is not significantly different between El Niño and La Niña (Fig. 1), independent of the types of El

[1]School of Oceanography, Shanghai Jiao Tong University, Shanghai, China. [2]Southern Marine Science and Engineering Guangdong Laboratory (Zhuhai), Zhuhai, China. [3]State Key Laboratory of Tropical Oceanography, South China Sea Institute of Oceanology, Chinese Academy of Sciences, Guangzhou, China. [4]Department of Atmospheric Sciences, National Taiwan University, Taipei, Taiwan. [5]Department of Atmospheric and Oceanic Science, University of Maryland, College Park, MD, USA. [6]Indian Institute of Technology, Bombay, Mumbai, India. ✉e-mail: zhoulei1588@sjtu.edu.cn

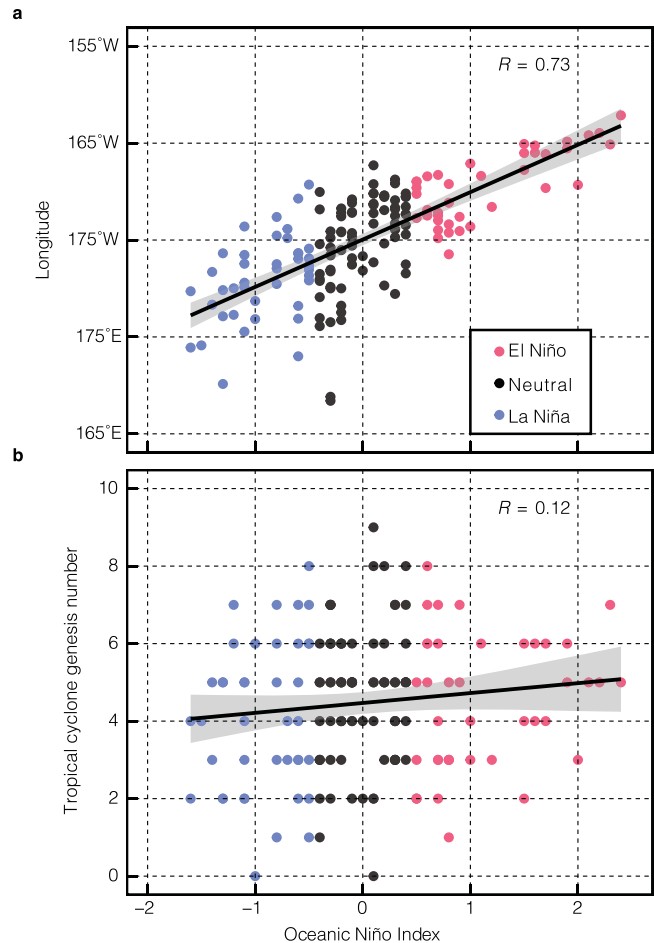

**Fig. 1 | The western Pacific warm pool expands during El Niño, but the tropical cyclone (TC) genesis number remains unchanged. a** The western Pacific warm pool centroid longitudes in El Niño, neutral, and La Niña phases are denoted by red, black, and blue filled circles, respectively. **b** same as **a** but for TC genesis numbers. El Niño and La Niña are denoted with the Oceanic Niño Index (see Methods). The western Pacific warm pool is defined as the region where sea surface temperatures are warmer than 28 °C. The TC genesis number is obtained from the Joint Typhoon Warning Center. The 95% confidence interval of the trend-lines are depicted in gray.

Niño[21,22]. Besides SST, subsurface ocean heat content[23,24] has also been suggested to be potentially influential for TC genesis[25]. Particularly, the heat content in the upper ocean, which is usually denoted with the depth of 26 °C isotherm, determines the available energy that can support TC evolution. Gray[26] listed "ocean thermal energy" as one of the six key parameters for TC formation.

In this study, we investigate the impacts of subsurface oceanic processes on the TC genesis, by contrasting El Niño and La Niña. During El Niño, the Ekman pumping due to surface wind anomalies shoals the depth of 26 °C isotherm and reduces the heat content in the upper ocean. As a result, the subsurface oceanic processes modify the favorable environment for TC genesis by reducing the warm water volume in the tropical central-north Pacific even as the SSTs seem to turn favorable and thus play an important role in TC genesis when an El Niño occurs. Our results suggest that if without the counter-acting effect from the ocean subsurface, TC frequency is expected to increase more, whereas because of the subsurface effect, the increase is less.

## Results

### Impacts of ocean subsurface on TC genesis

The thermodynamics of TCs is ideally modeled as a Carnot heat engine, running between a warm reservoir (the ocean) and a cold reservoir (the troposphere)[1,27]. Hence, the widely used genesis

potential indices[28,29] rely on one oceanic variable, i.e., SST. To include the possible contribution from ocean subsurface and quantify the influence of subsurface ocean heat, a TC genesis potential index (hereafter referred to as GPI$_{ocean}$) has been proposed for the western North Pacific TCs[30], i.e.,

$$\text{GPI}_{ocean} = p|10^5\eta_{1000}|^f \left(\frac{\bar{T}}{26}\right)^g \left(\frac{F}{45}\right)^h \left(\frac{D_{26}}{80}\right)^i \qquad (1)$$

where $\eta_{1000}$ is the absolute vorticity at 1000 hPa; $\bar{T}$ is the mean temperature in the upper mixed layer and the mixed layer bottom is the depth where temperature decreases by 0.2 °C from the temperature at the reference depth of 5 m[31]; $F$ is the net longwave radiation at the sea surface; $D_{26}$ is the depth of 26 °C isotherm; $f,g,h,i$ are constant coefficients; and $p$ is a coefficient which enables the best fit of GPI$_{ocean}$ to observations. The dependence of TC numbers and $D_{26}$ is shown in Supplementary Fig. S1 with observations. A strong vertical wind shear is usually unfavorable for TC genesis. However, it is generally weaker than 10 m s$^{-1}$ over most of the northwestern Pacific Ocean[32] and thus generally not large enough to prohibit TC genesis. Indeed, vertical wind shear was tested but GPI$_{ocean}$ was not found to be as sensitive to it as the factors listed above and was not retained in the GPI$_{ocean}$ calculation. The suitability of GPI$_{ocean}$ over other GPIs formulation for the western North Pacific TCs is that the ocean heat content is explicitly represented in terms of the depth of 26 °C isotherm, which facilitates the quantitative analyses in our study.

The TC genesis numbers during the peak typhoon season (from July to October) estimated with GPI$_{ocean}$ are shown in Fig. 2a–c, which agree with the best-track dataset developed by the U.S. Joint Typhoon Warning Center (JTWC; see Fig. 2d–f). Both observations and GPI$_{ocean}$ show that more than 90% of TCs originate to the west of 160°E during La Niña, while TC genesis locations extend eastward to about 170°E during El Niño albeit with a similar number of total TCs over the tropical western North Pacific. The performance of GPI$_{ocean}$ is consistent with GPIs defined by (28) and (29), as shown in Supplementary Fig. S2. The total change of GPI$_{ocean}$ (△GPI) between El Niño and La Niña can be divided into contributions of each variable as follows:

$$\triangle \text{GPI} = \frac{\partial \text{GPI}}{\partial \eta_{1000}} \bullet \triangle \eta_{1000} + \frac{\partial \text{GPI}}{\partial \bar{T}} \bullet \triangle \bar{T} + \frac{\partial \text{GPI}}{\partial F} \bullet \triangle F + \frac{\partial \text{GPI}}{\partial D_{26}} \bullet \triangle D_{26} \qquad (2)$$

The partial dependency of GPI$_{ocean}$ on each variable (e.g., $\frac{\partial \text{GPI}}{\partial D_{26}}$) is calculated using the climatology of all variables. The change in each variable (e.g., $\Delta D_{26}$) represents its difference between El Niño and La Niña. Particularly, the relative importance of $\bar{T}$ and $D_{26}$, both of which denote oceanic properties, can be estimated by comparing $\frac{\partial \text{GPI}}{\partial \bar{T}} \bullet \triangle \bar{T}$ and $\frac{\partial \text{GPI}}{\partial D_{26}} \bullet \triangle D_{26}$. This method is essentially the same as the one applied in (19), in which only one factor varies at a time while all other variables are held to their climatology.

As illustrated in Fig. 3a, b, in the tropical northwestern Pacific (dashed box within 5°N–20°N and 130°E–160°E), both $\bar{T}$ and $D_{26}$ have negative anomalies during El Niño, due to the weakening of easterly trade winds and eastward expansion of warm waters[33]. In the tropical central-north Pacific (solid box within 5°N–20°N and 160°E–170°W), it has been well documented and understood that $\bar{T}$ is warmer during El Niño than during La Niña (Fig. 3a). However, negative $D_{26}$ anomalies are evident beneath the positive $\bar{T}$ anomalies in the solid box during El Niño. Such patterns of $\bar{T}$ and $D_{26}$ can also be seen in the BOA_Argo data (Supplementary Fig. S3). The four variables incorporated into GPI$_{ocean}$ (Eq. 1) have commensurate contributions to the total change of GPI$_{ocean}$. During El Niño, the $\eta_{1000}$ and $F$ anomalies tend to increase TC genesis across the entire tropical Pacific (Supplementary Fig. S4). In the tropical northwestern Pacific (dashed box in Fig. 3), cold $\bar{T}$ and shallow $D_{26}$ anomalies jointly drive a moderate reduction of TC genesis

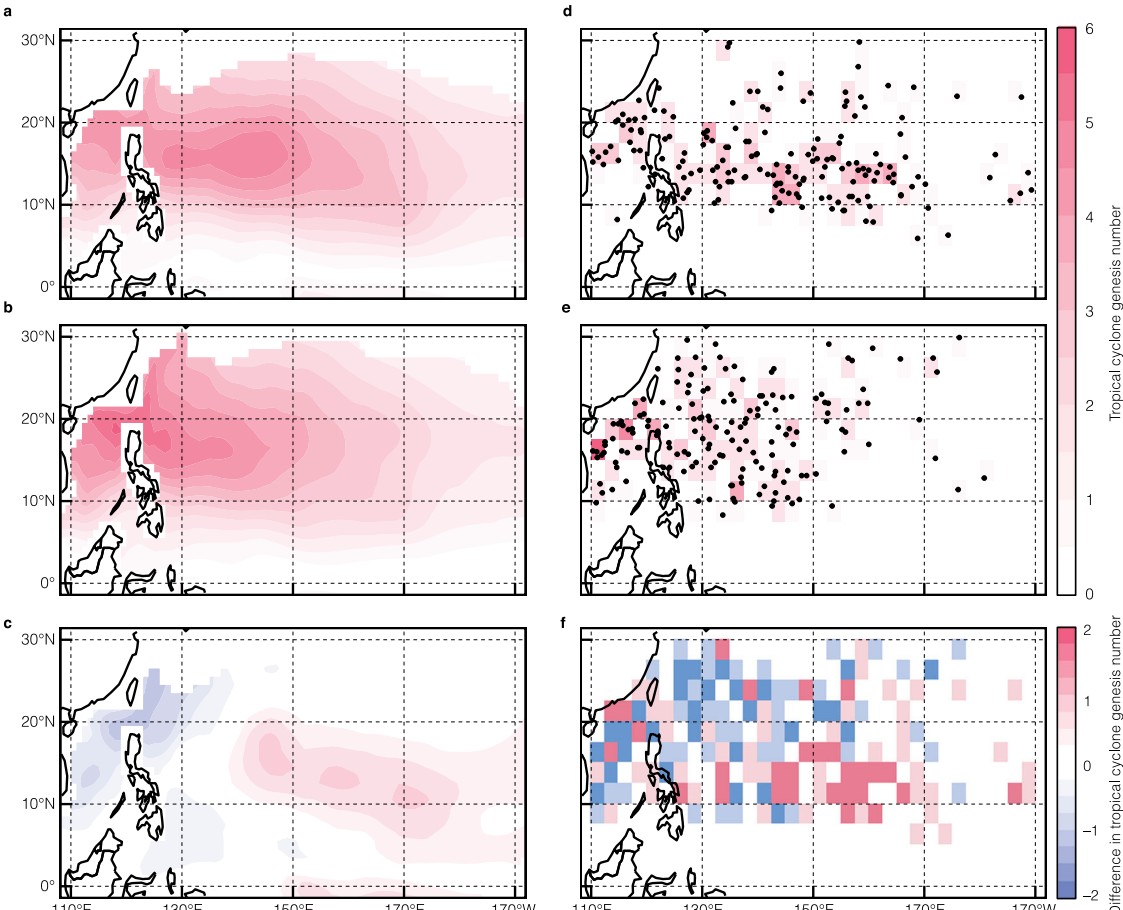

**Fig. 2 | Tropical cyclone (TC) genesis numbers during El Niño and La Niña and their differences. a** The TC genesis numbers estimated using the genesis potential index ($GPI_{ocean}$) during El Niño, **b** is for La Niña, and **c** is for the differences between El Niño and La Niña. **d–f** are the same as **a–c** but from the Joint Typhoon Warning Center. TC genesis numbers are binned to a grid of 2.5° longitude × 2.5° latitude. El Niño and La Niña are defined with the Oceanic Niño Index (see Methods) and are listed in Supplementary Table S1.

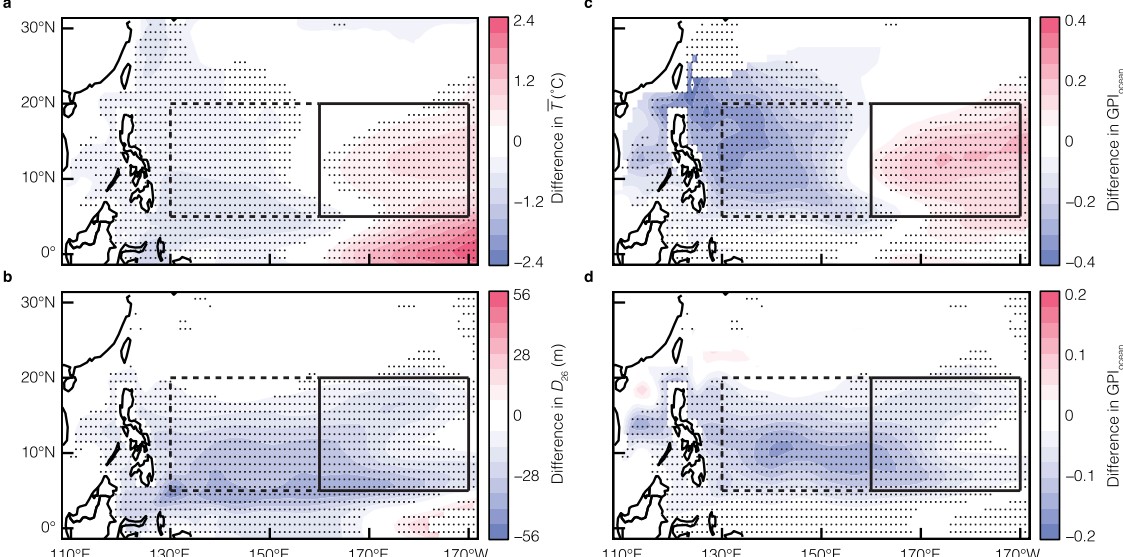

**Fig. 3 | Differences of the upper mixed layer ($\bar{T}$) and 26 °C isotherm depth ($D_{26}$) between the two phases of ENSO and their impacts on tropical cyclone genesis (GPI).** Differences of $\bar{T}$ (**a**) and $D_{26}$ (**b**) between El Niño and La Niña. The unit is °C for $\bar{T}$ and m for $D_{26}$. **c** $\frac{\partial GPI}{\partial \bar{T}} \cdot \Delta \bar{T}$, where $\Delta \bar{T}$ is the difference in mean $\bar{T}$ between El Niño and La Niña. **d** $\frac{\partial GPI}{\partial D_{26}} \cdot \Delta D_{26}$, where $\Delta D_{26}$ is the differences of $D_{26}$ between El Niño and La Niña; this represents the sensitivity of GPI to $D_{26}$. The black dots indicate the differences that are statistically significant at the 95% confidence level.

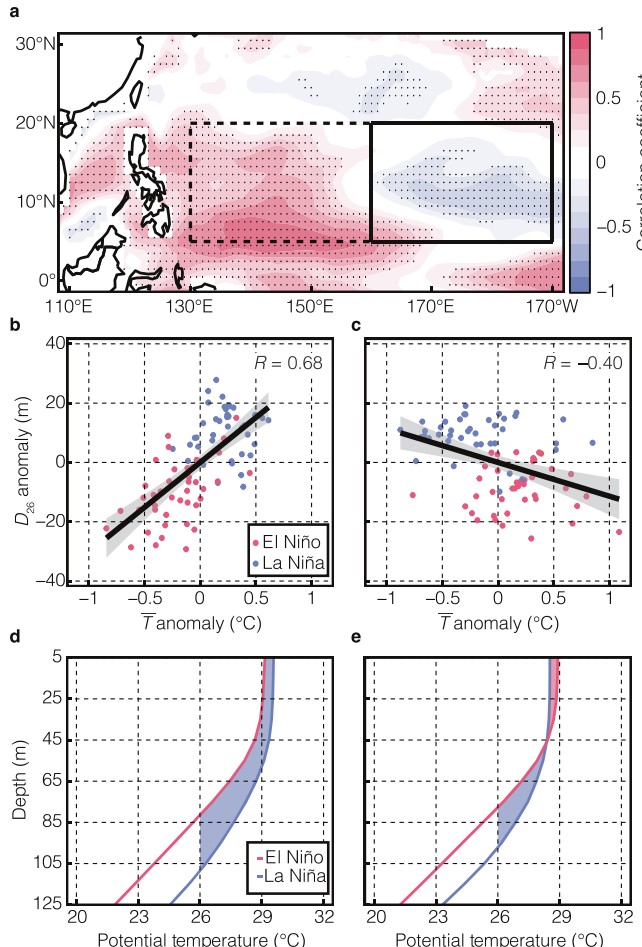

**Fig. 4 | Opposite correlations between the upper mixed layer ($\bar{T}$) and 26 °C isotherm depth ($D_{26}$) in the tropical northwestern Pacific and the tropical central-north Pacific. a** Correlation coefficients between $\bar{T}$ and $D_{26}$ during all months listed in Supplementary Table S1. The black dots indicate that the correlations are statistically significant at the 95% confidence level. **b** The scatter plot of mean $\bar{T}$ anomalies and $D_{26}$ anomalies averaged in the tropical northwestern Pacific (the dashed box). The red and blue dots indicate El Niño and La Niña, respectively. **c** Same as **b**, but for the tropical central-north Pacific (the solid box). The 95% confidence interval of the trendline is depicted in gray. **d** Vertical profiles of potential temperature for the dashed box. **e** Same as **d** but for the solid box. The shaded regions delineate the difference of upper layer heat content between El Niño and La Niña. The blue regions indicate the upper layer heat content is lower during El Niño, while the red region indicates the upper layer heat content is higher during El Niño.

during El Niño. In contrast, in the tropical central-north Pacific, the anomalies of all variables favor an increase in TC genesis, except for $D_{26}$. According to GPI$_{ocean}$, there are 0.15 more TCs (statistically significant at the 99% confidence level) generated per month in the tropical central-north Pacific (solid box in Fig. 3) during El Niño relative to La Niña. If $D_{26}$ was held to its climatology, there would be 0.19 more TCs (also significant at the 99% confidence level) generated per month during El Niño in the tropical central-north Pacific, which is a 27% increase in TC genesis. The conclusion remains valid (Supplementary Fig. S5), even when the El Niño events are categorized further into the central-Pacific (CP) and the eastern-Pacific (EP) types[5,34].

### Negative correlation between D$_{26}$ and SST in central-north Pacific

It is well known that the thermocline, traditionally approximated by the 20 °C isotherm[35], has a positive correlation with the SSTs[36]. The

positive correlation also applies to $D_{26}$, since the accumulation (dissipation) of warm water in the western Pacific deepens (shoals) the 26 °C isotherm depth (also seen as sea level height changes). This is confirmed in the tropical northwestern Pacific (dashed box in Fig. 4a), where the correlation coefficient between regional mean $\bar{T}$ and $D_{26}$ is 0.68 (Fig. 4b; significant at 99% confidence level). This positive correlation was used in many seminal studies[16,36] and served as a basic dynamic paradigm for ENSO studies. In addition, the positive correlation between $D_{26}$ and SST guarantees a positive correlation between SST and the heat content in the upper layer ($c_p\rho\int_{D_{26}}^{0}Tdz$ where $c_p$ is the heat capacity of seawater, $\rho$ is the density of seawater, $T$ is the potential temperature of seawater; Fig. 4d). However, in the tropical central-north Pacific (solid box in Fig. 4a), the situation is different from the classical understanding described above. $\bar{T}$ and $D_{26}$ have a significant negative correlation and the correlation coefficient is −0.40 (Fig. 4c). During El Niño, $D_{26}$ shoals while the SST increases. The former tends to reduce the heat content, while the latter favors the increase of heat content. As a result of these conflicting impacts, the relation between upper layer heat content and the SST anomalies becomes statistically insignificant (Fig. 4e). Therefore, the reduced heat content as well as the shallow $D_{26}$ in the tropical central Pacific result in an unexpected limitation on the TC genesis suggesting a delicate yet competing control between SST and the subsurface heat content in jointly modulating TC genesis.

The traditional positive correlation between SST anomalies and the 26 °C isotherm depth exists over much of the tropical oceans, which is dynamically established by the first baroclinic mode in the ocean[37]. Nevertheless, the tropical central-north Pacific (solid box in Fig. 4a) during ENSO is an exception. The Ekman pumping due to surface wind stress curl anomalies dominate the changes in $D_{26}$. During El Niño, the pronounced westerly wind anomalies in the deep tropics diminish meridionally (approximately from 5°N to 15°N; arrows in Fig. 5a). Therefore, the surface wind stress curl has cyclonic anomalies throughout the tropical northwestern and central-north Pacific (shading in Fig. 5a). The resulting Ekman suction is shown in Fig. 5b. The same conclusion applies to La Niña but in opposite direction, i.e., the anticyclonic wind stress curl results in a deeper $D_{26}$ across the tropical northwestern and central-north Pacific. The conspicuous negative correlations between spatial mean Ekman upwelling velocity induced by wind stress curl and $D_{26}$ are statistically significant as illustrated in Fig. 5c, d. Such wind stress curl anomalies are coherent with the variation of the North Pacific Subtropical High (NPSH; Supplementary Fig. S6), i.e., the weakening of the NPSH during El Niño[38] leads to a relaxation of tropical easterlies and reinforces the cyclonic wind stress curl anomalies in the tropics.

## Discussion

The positive correlation between $\bar{T}$ and $D_{26}$ in the tropical northwestern Pacific follows the canonical ENSO theories. However, in the north-central tropical Pacific, $\bar{T}$ increases due to the weakening of easterly winds and the consequent expansion of the Pacific warm pool. Meanwhile, $D_{26}$ shoals due to the anomalous Ekman suction associated with cyclonic wind stress curl which occurs over the NPSH during El Niño. Consequently, $\bar{T}$ and $D_{26}$ have a significantly negative correlation in the tropical central-north Pacific. The GPI offset by $D_{26}$ in the tropical central-north Pacific should not be neglected since TCs formed over the tropical central Pacific travel over a longer distance before landfall and thus has the potential to achieve a higher intensity with more devasting effects[18]. Such conclusions are also supported by the simulations from the High Resolution Model Intercomparison Project (HighResMIP; Supplementary Table S2 and Fig. S7a)[39].

It should be emphasized that there is no evidence of a difference in these processes due to different ENSO types[40]. Overall, although the western Pacific warm pool expands eastward considerably during El

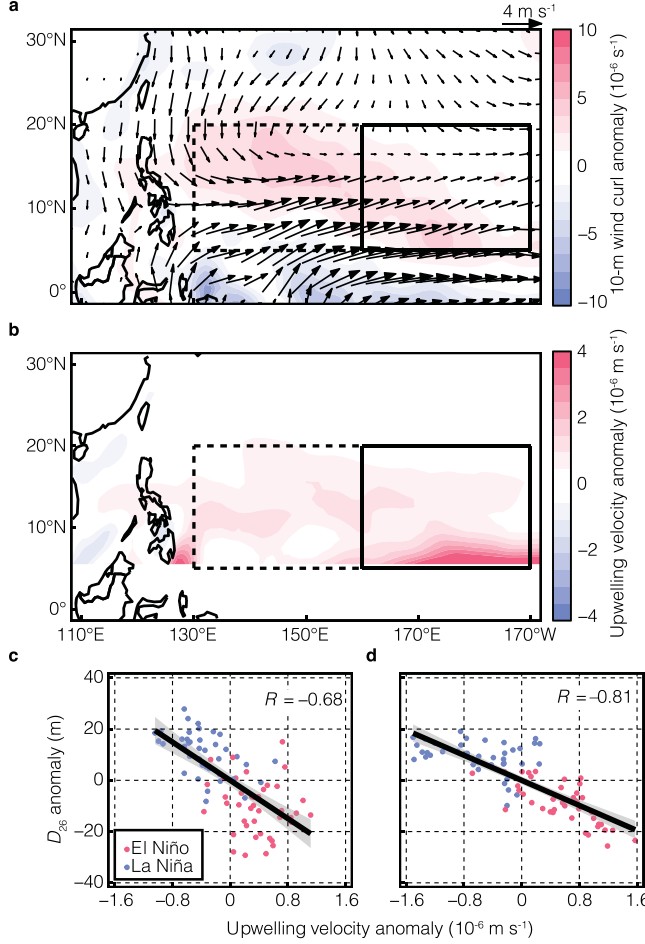

**Fig. 5 | Shoaling of the 26 °C isotherm depth ($D_{26}$) due to surface wind stress curl anomaly and its Ekman effect. a** Westerly wind anomalies (vector; unit: m s⁻¹) and 10-m wind curl anomalies (color shading; unit: $10^{-6}$ s⁻¹) during El Niño with respect to La Niña. **b** Ekman upwelling velocity anomalies during El Niño against La Niña and the unit is $10^{-6}$ m s⁻¹. **c** The scatter plot of regional mean Ekman upwelling velocity anomalies and $D_{26}$ anomalies in the tropical northwestern Pacific. The red (blue) dots are for El Niño (La Niña). **d** Same as **c** but for the tropical central-north Pacific. Both correlation coefficients are statistically significant at 99% confidence level and the 95% confidence interval of the trendline is depicted in gray.

Niño, due to a shoaling $D_{26}$ and the anti-correlated behavior between SST and the heat content in the tropical central-north Pacific, the TC genesis numbers do not increase by as many as expected from SST alone. Our results underscore the complexity of TC genesis under global warming, especially since many climate models project an El Niño-like pattern under global warming[3]. Therefore, the ocean sub-surface (vertical structure changes) and the SST changes must be considered in total for TC genesis in a warming world. ENSO impact on TC genesis provides an excellent analog and a cautionary tale for a potential cancellation of the dynamic and thermodynamic impacts on GPI.

A caveat here is that there are uncertainties which are expected to be eliminated by further analysis. For example, the main results have a quantitative dependence on the form of GPI (Supplementary Figs. S7 and S8), although the conclusions remain valid. Our results also do not exclude the possibility that intense typhoons (category 3 and above) may increase in number, since the accumulated cyclone energy (ACE) during El Niño exceeds that during La Niña (Supplementary Fig. S9) and more TCs can have a longer lifetime to grow over the warm ocean[41] after being born over the tropical central Pacific. However, the importance of the need to consider the combined effects of SST and

ocean heat content on TC genesis can hardly be overemphasized, especially in a warming world.

## Methods
### Data
The TC genesis data are obtained from the best-track dataset developed by the U.S. Joint Typhoon Warning Center (JTWC). In this study, a TC is generated when the maximum sustained wind speed reaches 34 knots for the first time. The TCs generated over the North Pacific which travel westward and threaten eastern Asia are considered in this study.

For the ocean variables, monthly ocean temperatures are obtained from the Global Ocean Data Assimilation System (GODAS) data products[42] developed by the National Centers for Environmental Prediction (NCEP). The horizontal resolution is 1/3° latitude × 1° longitude. The Simple Ocean Data Assimilation (SODA) version 3 reanalysis products[43] and the monthly Extended Reconstructed SST (ERSST) version 5 dataset[44] are also used. The results using SODA reanalysis and ERSST are qualitatively the same as the ones using GODAS. In addition, observations from Argo, i.e., BOA_Argo[45] are used to verify the results obtained from the analysis and reanalysis products.

For the atmosphere variables, wind velocities, specific humidity, air temperatures and surface heat fluxes are obtained from the monthly National Center for Environmental Prediction-National Center for Atmospheric Research (NCEP/NCAR) Reanalysis I products[46] with a resolution of 2.5° latitude × 2.5° longitude. The NCEP-DOE Reanalysis II products[47] and ERA5 reanalysis[48] products at European Centre for Medium-Range Weather Forecasts (ECMWF) are also applied.

**Definition of El Niño and La Niña.** The Oceanic Niño Index[49] (ONI) is used to define El Niño and La Niña. It is calculated as monthly SST anomalies in the Niño 3.4 region (5°S–5°N and 120°W-170°W) after a 3-month running mean. When the monthly ONI is higher (lower) than 0.5 °C (−0.5 °C) for at least 5 consecutive months, an El Niño (La Niña) event is defined. The peak typhoon season is from July to October. All months in the peak season that fall into the El Niño and La Niña events are listed in Table S1.

### Definition of CP- and EP-type El Niños
Based on the definition of El Niño, when the largest SST anomaly occurs to the east (west) of 150°W, the El Niño event is defined as an EP (CP) El Niño. The months for the CP- and EP-type El Niño during the peak season is classified in Supplementary Table S1.

### TC genesis potential index (GPI)
GPI is a statistical proxy for the TC dynamics in nature. It has been widely used to quantify the influences of various physical drivers on TCs[50–53]. Although the GPI is empirical and not based on the dynamical and physical constraints, it captures the intrinsic dynamical constraints during TCs and often out-performs the dynamical approaches[54–56]. The conclusions in this study are not sensitive to the specific form of GPI$_{ocean}$. Particularly, another GPI$_{atm\_ocean}$ was proposed in Eq. 4 in (30), which adopted $D_{26}$ as well as other atmospheric and oceanic variables. The Supplementary Fig. S8 shows the impacts of $D_{26}$ on TC genesis using GPI$_{atm\_ocean}$, which is qualitatively consistent with Fig. 3d, albeit with some quantitative differences. The quantitative differences between GPI$_{ocean}$ and GPI$_{atm\_ocean}$ are reproduced with HighResMIP outputs and the same conclusion can be drawn from them as well (Supplementary Fig. S7).

### Ekman pumping/suction
The Ekman pumping and suction is calculated as

$$w_e = \frac{1}{\rho}\left(\frac{\partial M_x}{\partial x} + \frac{\partial M_y}{\partial y}\right) \tag{3}$$

$$M_x = \frac{\tau_y}{f} \quad (4)$$

$$M_y = -\frac{\tau_x}{f} \quad (5)$$

where $\rho$ is seawater density; $M_x$ and $M_y$ are zonal and meridional Ekman mass transports, respectively; $\tau_x$ and $\tau_y$ are zonal and meridional 10-m wind stresses, respectively; and $f$ is Coriolis parameter.

## Data availability

All datasets used in this study are publicly available. JTWC data are available from the Joint Typhoon Warning Center (https://www.metoc.navy.mil/jtwc/jtwc.html?best-tracks); GODAS data are available from Physical Sciences Laboratory (http://www.esrl.noaa.gov/psd/data/gridded/data.godas.html); SODA data are available from http://www.soda.umd.edu/; ERSST data are available from National Centers for Environmental Information (https://www.ncei.noaa.gov/products/extended-reconstructed-sst); BOA_Argo data are available from China Argo Real-time Data Center (http://www.argo.org.cn/index.php?m=content&c=index&a=lists&catid=101); NCEP/NCAR Reanalysis I data are available from Physical Sciences Laboratory (https://psl.noaa.gov/data/gridded/data.ncep.reanalysis.html); NCEP-DOE Reanalysis II are available from Physical Sciences Laboratory (https://psl.noaa.gov/data/gridded/data.ncep.reanalysis2.html); ERA5 data are available from ECMWF (https://www.ecmwf.int/en/forecasts/datasets/reanalysis-datasets/era5); HighResMIP model outputs are available from the Earth System Grid Federation (ESGF; https://esgf-index1.ceda.ac.uk/search/cmip6-ceda/).

## Code availability

The computer codes used to analyze the data are available from the corresponding author on request.

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

## Acknowledgements

C.G. and L.Z. are supported by grants from the National Natural Science Foundation of China (42125601 and 42076001), Innovation Group Project of Southern Marine Science and Engineering Guangdong Laboratory (Zhuhai) (311020004), and the Oceanic Interdisciplinary Program of Shanghai Jiao Tong University (SL2020PT205). C.W. is supported by the National Natural Science Foundation of China (42192564), the National Key R&D Program of China (2019YFA0606701), the Strategic Priority Research Program of Chinese Academy of Sciences (XDB42000000 and XDA20060502), and Key Special Project for Introduced Talents Team of Southern Marine Science and Engineering Guangdong Laboratory (Guangzhou) (GML2019ZD0306). R.M. gratefully acknowledges the CYGNSS grant from NASA and the National Monsoon Mission funds for partial support. R.M. gratefully acknowledges the Visiting Faculty position at the Indian Institute of Technology, Bombay.

## Author contributions

C.G., L.Z., C.W., I.L., and R.M. conceived the central idea. C.G. and L.Z. performed the analyses and generated the figures. L.Z. and C.G. wrote the main paper with further inputs and edits from C.W., I.L., and R.M. C.G., L.Z., C.W., I.L., and R.M. contributed to the discussion of the results and commented on the paper.

## Competing interests

The authors declare no competing interests.
