## [Peer Review File · Nature Communications]

Unexpected Limitation of Tropical Cyclone Genesis by Subsurface Tropical Central-North Pacific during El NiñoReviewers' comments:

Reviewer #1 (Remarks to the Author):

The author used a data fitted empirical function G for TC genesis to infer the role of subsurface temperature measured by the depth (D26) of T=26C surface. I found some issues in their approach and reasoning to reach their conclusions. Thereby I cannot recommend this paper for considerations for publication in NC at this point.

#1 On the problem with G function derivation and inferences of contributions from D26. Unlike the formulation of maximum potential intensity (MPI) of TC, the G function and its dependence on a bunch of variables in all three papers (26, 27, 28) were empirical and not based on the solid dynamical and physical constraints. Murakami and Wang (ref. #27) found the original formula by K. A. Emanuel, D. S. Nolan (2004, ref #26) was problematic, although it has been used in the research community by many. M. Zhang, L. Zhou, D. Chen, C. Wang (ref # 28) further developed two new forms for G functions, trying to combine different combinations for atmospheric and ocean variables to fit the TC genesis data. The four forms of G functions of these three papers are all very different, all work well or to some extent in terms of fitting. But, this is a "big" problem for attributions to tell apart about what are factors "cause" the variations in G! Here I give a mathematical proof to demonstrate that using the two different forms of G function in Ref #2 which has two co-authors appear to be the co-authors of this manuscript as well. One is so-called oceanic G function which used in this manuscript, I will denote it as G-o. The other they referred it as atmospheric and oceanic G function, denoted G-ao. These two G functions are quite different but have shared considerations for the dependences of D26.

Let me simply rewrite them into the following form:

$$G=G_o=F_1 X^i, G=G_{ao}=F_2 X^j$$

Here $X=D26/80$ is what they have used. F_1 and F_2 are different functions which depend on a bunch of different variables and can be found from equation 3 and 4 of the ref#28. If we wish to attribute how variations in G are partitioned among changes due to ΔX and ΔF_1 , or ΔX and ΔF_2 , we have the following formulations

$$\Delta G=(iG)/(X) \Delta X + X^i \Delta F_1, \Delta G=(jG)/(X) \Delta X + X^j \Delta F_2$$

Here I used the fact $G=(F_1)X^i=(F_2)X^j$ by definition. In ref #27, they found that $i=0.25$, $j=0.13$. Please note the ΔX is the same and $G/(X)$ is the same, but two formulas will give almost a factor 2 difference in terms of contributions from D26 to ΔG . In other words, I give a mathematical proof here that the estimated contribution from D26 will be cut into half if the authors use the equation 4 instead of ref#28.

Mathematically, this problem of indeterminacy comes from the fact that predictors are not uncorrelated. But more fundamentally, it comes from the lack of physical constraints on the forms of G function, a problem not solved even by the best TC dynamist, like Prof. Emanuel. Anyone can propose his new versions by fitting the data, but the fact we have a rather small sample set of TC genesis data set will not give a solid answer until a breakthrough is made with a theory for GPI, which is not in sight.

Dynamically, I do not see a strong argument why a small change in D26 in the regions with 28.5-29C warm surface water as shown in Fig.4D&E will matter that much for TC genesis. How strongly a pre-TC depression can be affected by subsurface at 80m below? As it is unlikely a pre-TC depression can be strongly affected by the deep water, I am skeptical about this X factor for GPI. The indeterminacy as shown above using the results in ref#28 lends support to my point as well. My objection points to an essential weakness of the work.

#2 There is a shift of pattern TC genesis in the central to western north Pacific as shown in Fig.2, But this shift can be more likely due to shift in vertical wind-shear patterns rather than due to D26. Surface wind patterns shown in Fig.5A would indicate a change in vertical shear in the region about 5-15N and in the so-called green and yellow box which may have different contributions to G function variations. If the authors formulated the G function with considerations of vertical shear instead of D26, the changes in G function will be attributed to something else other than D26. Thus the inference drawn from G functional analysis are nonunique. The hypothesis about importance of D26 has to be further supported such as by CGCM simulations to demonstrate that D26 really matters.

#3 The authors tend to stress " The negative correlation between warm SST anomalies and

shallower 26C isotherm depth is opposite to the current understanding for ENSO dynamics" . But this has little to do with ENSO dynamics. The difference in green and yellow box in terms of t_{sub} and SST is more related to the difference in heat flux due to wind evaporations, which can become clear if one adds the wind pattern in Fig.5A onto the summer climatology wind, at least to my eyeballing estimations.

I do not rule out completely the role of subsurface ocean in GPI. But I provide my assessments to convince the authors that they need to provide direct evidence to prove (or falsify) the hypothesis that the D26 matters for GPI. The empirical G_s are not adequate and easily misleading as I can see into it. As I personally know the authors well, I am signing this negative but otherwise constructive review (with an interesting mathematical analysis which potentially can be a useful tool itself for the authors to use). **REVIEWER COMMENTS**

Reviewer #1

The authors addressed most of my concerns. I am happy to recommend for its publication.

I do have two more suggestions.

As I am especial happy to see they indeed dogged into some existing high resolutions simulations to provide some modeling evidence about the potential effect of D26 on TC genesis (Fig. R8), I wish they do so more directly. This figure is suggestive but can be not directly compared with the observed results in the key figure of this paper (Fig3D). They can use the model simulations and their choice of GPI formula to produce a similar figure (multi-model composite or etc) as Fig.3D. This kind of figures provide more direct evidence and informative addition if the results are as truly clearly supportive.

I also wish they do include another figure like Fig3D but used GPI_{oa} formulation as well. The difference in exponents of the two formulations yield a definite difference in terms of the level of D26 effect as I demonstrated mathematically. Whether it is a factor 2 or 1.5 (or somewhere in between) difference may depend on how one linearizing the GPI. But this point of mine is beyond question as it clearly shown in Fig.R2. I urge the authors to include this information in the extend figures so that readers will be aware with this caveat, which leaves door open for someone to go further to find ways to eliminate this uncertainty.

Reviewer #2

The author used a data fitted empirical function G for TC genesis to infer the role of subsurface temperature measured by the depth (D26) of T=26C surface. I found some serious issues in their approach and reasoning to reach their conclusions. Thereby I cannot recommend this paper for considerations for publication in NC.

#1 On the problem with G function derivation and inferences of contributions from D26.

Unlike the formulation of maximum potential intensity (MPI) of TC, the G function and its dependences on a bunch a variables in all three papers (26, 27, 28) were empirical and not based on the solid dynamical and physical constrains. Murakami and Wang (ref. #27) found the original formula by K. A. Emanuel, D. S. Nolan (2004, ref #26) was problematic, although it has been used in the research community by many. M. Zhang, L. Zhou, D. Chen, C. Wang (ref # 28) further developed two new forms for G functions, trying to combine different combinations for atmospheric and ocean variables to fit the TC genesis data. The four forms of G functions of these three paper are all very different, all works well or some extent in term of fitting. But, this is a "big" problem for attributions to tall apart about what are factors "cause" the variations in G!

Here I give a mathematical proof to demonstrate that using the two different forms of G function in Ref #2 which has two co-authors appear to be the co-authors of this manuscript as well. One is so-called oceanic G function which used in this manuscript, I will denote it as G-o. The other they refereed it as atmospheric and oceanic G function, denoted G-ao. These two G functions as quite different but have shared considerations for the dependences of D26.

Let me simply rewrite them into the following form:

$$G = G_o = F_1 X^i, \quad G = G_{ao} = F_2 X^j$$

Here $X=D_{26}/80$ is what they have used. F1 and F2 are different functions which depends on a bunch of different variables and can be found from equation 3 and 4 of the ref#28. If we wish to attribute how variations in G is partitioned among changes due to ΔX and $\Delta F1$, or ΔX and $\Delta F2$, we have the following formulations

$$\Delta G = \frac{i\bar{G}}{\bar{X}} \Delta X + \bar{X}^i \Delta F1, \quad \Delta G = \frac{j\bar{G}}{\bar{X}} \Delta X + \bar{X}^j \Delta F2$$

Here I used the fact $\bar{G} = \bar{F1}\bar{X}^i = \bar{F2}\bar{X}^j$ by *definition*. In ref #27, they found that $i=0.25$, $j=0.13$. Please note the ΔX is the same and $\frac{\bar{G}}{\bar{X}}$ is the same, but two formulas will give almost a factor 2 difference in terms of contributions from D26 to ΔG . In other words, I give a mathematical prove here that the estimated contribution from D26 will be cut into half if the authors use the equation 4 instead of ref#28.

Mathematically, this problem of indetermination comes from the fact that predictors are not uncorrelated. But more fundamentally, it comes from the lack of physical constrains on the forms of G function, a problem not solved even by the best TC dynamist, like Prof. Emanuel. Anyone can propose his new versions by fitting the data, but the fact we have a rather small

sample set of TC genesis data set will not give a solid answer until a breakthrough is made with a theory for GPI, which is not in sight.

Dynamically, I do not see a strong argument why a small changes in D_{26} in the regions with 28.5-29C warm surface water as shown in Fig.4D&E will matter that much for TC genesis. How strongly a pre-TC depression can be affected by subsurface at 80m bellow? As it is unlikely a pre-TC depression can be strongly affected by the deep water, I am skeptical about this X factor for GPI. The indeterminacy as shown above using the results in ref#28 lends support to my point as well. My objection points to an essential weakness of the work.

#2 There is a shift of pattern TC genesis in the central to western north Pacific as shown In Fig.2, But this shift can be more likely due to shift in vertical wind-shear patterns rather than due to D_{26} . Surface wind patterns shown in Fig.5A would indicate a change in vertical shear in the region about 5-15N and in the so-called green and yellow box which may have different contributions to G function variations. If the authors formulated the G function with considerations of vertical shear instead of D_{26} , the changes in G function will be attributed to something else other than D_{26} . Thus, the inference draw from G functional analysis are nonunique. The hypothesis about importance of D_{26} must be further supported such as by CGCM simulations to demonstrate that D_{26} really matters.

#3 The authors tend to stress " The negative correlation between warm SST anomalies and shallower 26C isotherm depth is opposite to the current understanding for ENSO dynamics". But this has little to do with ENSO dynamics. The difference in green and yellow box in terms of t_{sub} and SST is more related to the difference in heat flux due to wind evaporations, which can become clear if one adds the wind pattern in Fig.5A onto the summer climatology wind, at least to my eyeballing estimations.

I do not role out completely the role of subsurface ocean in GPI. But I provide my assessments to convince the authors that they need to provide direct evidence to prove (or falsify) the hypothesis that the D_{26} matters for GPI. The empirical Gs are not adequate and easily misleading as I can see into it. As I personally know the authors well, I am signing this negative but otherwise constructive review (with an interesting mathematical analysis which potentially can be a useful tool itself for the authors to use).

Fei-Fei Jin

Reviewer #2 (Remarks to the Author):

Review of

Suppression of Tropical Cyclone Genesis by Subsurface Environment in the Tropical Central-North Pacific during El Niño

by

Authors: Cong Gao¹, Lei Zhou^{1, 2*}, Chunzai Wang³, I.-I. Lin⁴, Raghu Murtugudde^{5, 6}

In this study, the authors highlight a counterintuitive relationship between the potential TC genesis in the Central North Pacific and the phase of ENSO. Whereas the warm El Niño phase is characterized by an extension of the warm pool and warmer surface anomalies in the Central-north Pacific, the depth of the isoT26C is actually shallower (due to increased Ekman suction in relation with the large scale wind stress anomalies associated with the North Pacific subtropical high) making the heat content lower and counteracting the favorable sea surface environment for TC activity which leads to a suppression of TC genesis in this region. While the idea and proposed mechanism are interesting, I think the conclusion is a bit overstated. The authors claimed a suppression, whereas it is merely a reduction of the TC genesis potential. This study is well written and has the potential for publication in Nature Communication but would require more convincing analysis or a shift in focus beforehand.

Major comments:

1/ Figure 2C displays actually an increase in TC genesis during El Niño phases compared to La Niña's. In addition, I wonder if the unchanged TC number between the 2 phases of ENSO shown in their figure 1 comes from the choice of their averaging region (not clearly specified by the way, but which, I assume, encompasses the whole warm pool) that integrates the reduction of TC number around the Philippines and the increase between 140 and 170W and lead to an overall similar number of TC during both ENSO phases.

What if this figure was done using the region delineated by their yellow dashed box, would the increase in TC genesis during El Niño remain insignificant?

2/ Maybe the authors could explore more the limitation in increase/reduction of potential TC genesis during El Niño/La Niña phases compared to climatological/neutral ENSO conditions? For instance, the figure 2 of Lin et al. (2020) shows a limited increase/reduction in TC activity during El Niño/La Niña in the Central North Pacific compared to the westernmost part of the basin that might come from the proposed competing mechanism between SST and heat content:

- A limitation in TC genesis in that region during El Niño despite positive SST anomalies due to the shoaling of D26
- the opposite during La Niña due to a reduction of SST anomalies but an increase in heat content

3/ Also, why not extending the analysis to the entire central Pacific (i.e. until 150°W, the usual definition of the Central North Pacific TC basin)?

4/ Knowing that the heat content is more influential on TC intensification than genesis, I wonder if these results would be changed (even maybe strengthened) if the authors used an index accounting for TC intensity (the Accumulated Cyclone Energy for instance, Bell et al 2000, 2004) as compared to an index of TC genesis or limited their analysis to the occurrence of major TC (Category 3 and above) in this region?

Minor comments:

Reference 13 is not the best suited

Line 42: I thought the term had been coined "hiatus"

Line 53: Phase not flavor

Reference:

- Bell, G. D., and Coauthors, 2000: Climate Assessment for 1999. *Bull. Amer. Meteor. Soc.*, 81 , S1–S50.
- Bell, G. D., S. Goldenberg, C. Landsea, E. Blake, R. Pasch, M. Chelliah, and K. Mo, 2004: The 2003 Atlantic hurricane season [in "State of the Climate in 2003"]. *Bull. Amer. Meteor. Soc.*, 85 .(6), S20–S24.
- Lin, I.-I., Camargo, S. J., Patricola, C. M., Boucharel, J., Chand, S., Klotzbach, P., Chan, J. C. L., Wang, B., Chang, P., Li, T., & Jin, F. F. (2020). ENSO and tropical cyclones. In M. J. McPhaden, A. Santoso & W. Cai (Eds.), *El Niño Southern Oscillation in a changing climate*, Geophysical Monograph Series (Chapter 17, pp. 337- 408). Hoboken, NJ and Washington DC, USA: John Wiley - Sons, Inc. and American Geophysical Union.

Responses to Reviewers

We greatly appreciate the insightful comments, and the very specific and valuable suggestions provided by the two reviewers. The manuscript is revised accordingly, and we hope that the revisions are satisfactory to the reviewers.

5

Reviewer #1 (Remarks to the Author):

The author used a data fitted empirical function G for TC genesis to infer the role of subsurface temperature measured by the depth (D_{26}) of $T=26^{\circ}\text{C}$ surface. I found some issues in their approach and reasoning to reach their conclusions. Thereby I cannot recommend this paper for considerations for publication in NC at this point.

10

#1 On the problem with G function derivation and inferences of contributions from D_{26} .

Unlike the formulation of maximum potential intensity (MPI) of TC, the G function and its dependance on a bunch a variables in all three papers (26, 27, 28) were empirical and not based on the solid dynamical and physical constrains. Murakami and Wang (ref # 27) found the original formula by K. A. Emanuel, D. S. Nolan (2004, ref # 26) was problematic, although it has been used in the research community by many. M. Zhang, L. Zhou, D. Chen, C. Wang (ref # 28) further developed two new forms for G functions, trying to combine different combinations for atmospheric and ocean variables to fit the TC genesis data. The four forms of G functions of these three papers are all very different, all works well or some extent in terms of fitting. But, this is a "big" problem for attributions to tall apart about what are factors "cause" the variations in G !

15

20

Reply:

We agree with the reviewer that the GPIs “were empirical”. Nevertheless, nature is so complex that existing theories on the dynamical control on TC genesis are still not satisfactory. Otherwise, empirical methods such as GPIs would not have been developed, if the theories were adequate. The reviewer has also acknowledged that the GPIs, particularly the one by Emanuel and Nolan (2004), have been widely applied and many useful conclusions have been drawn (such as Emanuel 2013; Lavender et al. 2018; Zhao et al. 2020; Yang et al. 2021). Moreover, the GPIs have

25

30 also been used to quantify the influences of various physical features on tropical cyclones (TCs),
such as in Patricola et al. (2016), Cao et al. (2021), Fu et al. (2021), and Murakami (2022).
Therefore, although the dynamical and physical bases for GPIs are still a grand scientific challenge,
they have demonstrated applicability and usefulness for exploring the interactions between TCs
and the environment. We humbly submit that the science of improving GPIs and advancing the
35 process and predictive understandings of TCs based on both dynamical and statistical approaches
can proceed hand-in-hand and complementarily.

A well-known example is the use of Statistical Hurricane Intensity Prediction Scheme (SHIPS;
DeMaria and Kaplan 1994; DeMaria et al. 2005) and the Statistical Typhoon Intensity Prediction
Scheme (STIPS; Knaff et al. 2005) operationally by NOAA. TC is truly a complex system and
40 current understanding on its dynamics still has large room for improvement. The statistically based
schemes, such as SHIPS and STIPS, have often been found to out-perform the dynamical
approaches (<https://rammb2.cira.colostate.edu/research/tropical-cyclones/ships/>). Considering the
potential to save lives, infrastructure, and property, it is entirely justified to employ empirical
approaches till dynamical approaches eliminate them.

45 The new Columbia HAZard (CHAZ) model developed by the Columbia University is another
example (Lee et al. 2020). Though it is statistically based, it has wide applications in both global
warming projections and the present-climate applications. Though published only in 2020, it has
already ~40 citations. In the 35th and 34th American Meteorological Society (AMS)'s Hurricane
and Tropical Meteorology conferences (May 2022 and April 2021), this statistically based
50 approach also received much attention and was discussed extensively.

It is also true that “Murakami and Wang (ref # 27) found the original formula by Emanuel
and Nolan (2004, ref # 26) was problematic”. Murakami and Wang (2010) pointed out that the
“upward motion was not fully taken into account in the original GPI [the one in Emanuel and
Nolan (2004)]”. They made an improvement “by explicitly incorporating the following vertical
55 motion term” and proposed another GPI in Eq. (2) in their paper. The modification in Murakami
and Wang (2010) was also made on an empirical basis, rather than a solid dynamical basis.
However, the results in Murakami and Wang (2010) indicated that “the impact of remote
dynamical forcing (on TCs) is greater than that of local thermodynamical forcing” in the western
North Atlantic. Hence, this is a good example that an empirical GPI promoted our understandings
60 in the environmental impacts on TCs.

In addition, although GPI_o proposed in Zhang et al. (2016) presented the empirical relations between the TC genesis numbers and different environmental variables, the variables were not selected randomly or artificially. The details of the selection are:

- 65 1. All variables were selected based on previous process studies, which had already unveiled the dynamical relations between each variable and TCs.
- 70 2. GPI_o was obtained with a recursive regression algorithm. The flowchart is shown in Fig. R1. The algorithm guaranteed that, statistically, (1) the inclusion of any new variable from the candidate pool cannot improve GPI_o ; (2) the removal of any variable from GPI_o would significantly deteriorate the performance of GPI_o . Therefore, GPI_o does have an implicit dynamical basis and is not totally empirical. It is unique, as long as the large candidate pool remains the same. See more technical details in Zhang et al. (2016).
- 75 3. The GPIs are a statistical proxy for the TC dynamics in nature. Although no dynamics were considered explicitly when creating GPIs, all GPIs reflect the dynamical constraints which are intrinsic to the TC system and life cycle. That is probably why, as the reviewer stated, “all (four GPIs) work well to some extent in terms of fitting”. In Murakami and Wang (2010), it was also claimed that the “upward motion was not fully taken into account in the original GPI, although the relative humidity term may reflect it to some degree”. In the response to a following comment, we will show that the same conclusions about the impacts of D_{26} on TC genesis numbers can be drawn when using both GPI_o and GPI_{ao} , which have different forms.

80

Overall, an item f is added in the Methods section in the revised manuscript, which briefly summarizes the above explanations.

Figure R1 Flowchart of the regression algorithm which was used to obtain GPI_o in Zhang et al. (2016). It begins with a null model (no variables). When a candidate variable X_i is added to the GPI (denoted with Y), an F -test is performed to determine whether the inclusion would significantly improve the GPI, i.e., increase the match between GPI and observations. If yes, the variable X_i is kept in GPI, otherwise it is removed. Note that for each step, the F -test is not only performed to the new variable X_i but the test is also performed to all variables which are already selected. As a result, any variable which could not contribute significantly to the improvement of GPI would be returned to the candidate pool. This is a recursive process. The regression process stops when no new variable can be included into the GPI and no variable can be removed from the GPI. Therefore, the final GPI does not depend on the order of feeding the variables into the regression system.

#2 Here I give a mathematical proof to demonstrate that using the two different forms of G function in Ref #27 which has two co-authors appear to be the co-authors of this manuscript as well. One is so-called oceanic G function which used in this manuscript, I will denote it as G_o . The other they referred it as atmospheric and oceanic G function, denoted G_{ao} . These two G functions as quite different but have shared considerations for the dependences of D_{26} .

Let me simply rewrite them into the following form:

$$G = G_o = F_1 X^i, G = G_{ao} = F_2 X^j$$

105 Here $X = D_{26} / 80$ is what they have used. F_1 and F_2 are different functions which depends on a bunch of different variables and can be found from equation 3 and 4 of the ref#28. If we wish to attribute how variations in G is partitioned among changes due to ΔX and ΔF_1 , or ΔX and ΔF_2 , we have the following formulations

$$\Delta G = \frac{i\bar{G}}{\bar{X}} \Delta X + \bar{X}^i \Delta F_1, \Delta G = \frac{j\bar{G}}{\bar{X}} \Delta X + \bar{X}^j \Delta F_2$$

110 Here I used the fact $\bar{G} = \bar{F}_1 \bar{X}^i = \bar{F}_2 \bar{X}^j$ by definition. In ref # 27, they found that $i=0.25, j=0.13$. Please note the ΔX is the same and $\frac{\bar{G}}{\bar{X}}$ is the same, but two formulas will give almost a factor 2 difference in terms of contributions from D_{26} to ΔG . In other words, I give a mathematical prove here that the estimated contribution from D_{26} will be cut into half if the authors use the equation 4 instead of ref # 28.

115 Mathematically, this problem of indetermination comes from the fact that predictors are not uncorrelated. But more fundamentally, it comes from the lack of physical constrains on the forms of G function, a problem not solved even by the best TC dynamist, like Prof. Emanuel. Anyone can propose his new versions by fitting the data, but the fact we have a rather small sample set of TC genesis data set will not give a solid answer until a breakthrough is made with a theory for GPI, which is not in sight.

Reply:

120 We appreciate the reviewer's effort in providing a detailed mathematical proof to verify our results. To the best of our understanding, the key concern that the reviewer raised is whether our current conclusions depend on the specific form of the GPI. With the following analyses, it is shown that the two different GPIs in Zhang et al. (2016) yield consistent conclusions.

125 Firstly, we agree with the reviewer that the variables in any GPI are not independent from each other. This was clarified in Murakami and Wang (2010) as well, as noted in the above response. As a result, the specific form of a variable (such as D_{26}) may change in different GPIs, after different variables are selected. This is also a reflection of the intrinsic inter-dependence of different variables. One can create a GPI with orthogonal variables by applying the Gram-Schmidt process (Ford 2015), but it would render difficult the physical interpretations.

130 Secondly, we found that the mathematical processes in the comment need to be modified.
 Following the symbols used in the comment, one has

$$\Delta G_o \approx \frac{\partial \overline{G_o}}{\partial X} \Delta X + \frac{\partial \overline{G_o}}{\partial F_1} \Delta F_1 = \frac{\overline{iF_1}}{X^{1-i}} \Delta X + \overline{X^i} \Delta F_1 = \frac{\overline{iG_o}}{X} \Delta X + \overline{X^i} \Delta F_1,$$

$$\Delta G_{ao} \approx \frac{\partial \overline{G_{ao}}}{\partial X} \Delta X + \frac{\partial \overline{G_{ao}}}{\partial F_2} \Delta F_2 = \frac{\overline{jF_2}}{X^{1-j}} \Delta X + \overline{X^j} \Delta F_2 = \frac{\overline{jG_{ao}}}{X} \Delta X + \overline{X^j} \Delta F_2.$$

135 $\overline{X^i}$ is the climatological mean of X^i , which is different from \overline{X}^i (i.e., \overline{X} to the i th power). The
 difference between $\overline{G_o}$ and $\overline{G_{ao}}$ is neglected, which was also assumed by the reviewer. However,

$$\frac{\overline{iG_o}}{X} = \overline{iG_o \frac{1}{X}} \neq \overline{iG_o} \frac{1}{\overline{X}} \neq \frac{i\overline{G}}{\overline{X}},$$

$$\frac{\overline{jG_{ao}}}{X} = \overline{jG_{ao} \frac{1}{X}} \neq \overline{jG_{ao}} \frac{1}{\overline{X}} \neq \frac{j\overline{G}}{\overline{X}},$$

because G_o, G_{ao} are dependent on X . Hence, the following two equations in the comments, i.e.,

$$\Delta G_o = \frac{i\overline{G}}{\overline{X}} \Delta X + \overline{X^i} \Delta F_1,$$

140
$$\Delta G_{ao} = \frac{j\overline{G}}{\overline{X}} \Delta X + \overline{X^j} \Delta F_2,$$

cannot be strictly valid. Therefore, it cannot be concluded based on the above two equations that
 “the estimated contribution from D_{26} will be cut into half if the authors use the equation 4 in Zhang
 et al. (ref # 28)”.

145 Thirdly, we repeated the calculations with G_{ao} and reached similar conclusions about the
 influences of D_{26} on TC genesis numbers. The results are shown in Fig. R2B, which are consistent
 with Fig. R2A, i.e., Fig. 3D of the manuscript. There are quantitative differences, but the
 conclusions remain the same. This (the robust conclusion with quantitative differences) is
 mentioned in the new item f in the Methods section in the revised manuscript.

150 **Figure R2** The contribution of D_{26} in regulating TC genesis calculated (A) with G_o and (B) with
 G_{ao} in Zhang et al. (2016).

#3 Dynamically, I do not see a strong argument why a small change in D_{26} in the regions with 28.5-29C warm surface water as shown in Fig.4D&E will matter that much for TC genesis. How strongly a pre-TC depression can be affected by subsurface at 80m bellow? As it is unlikely a pre-TC depression can be strongly affected by the deep water, I am skeptical about this X factor for GPI. The indeterminacy as shown above using the results in ref#28 lends support to my point as well. My objection points to an essential weakness of the work.
Reply:

160 We appreciate the reviewer for raising the essential issue whether the variation at the 26°C isotherm can have an impact on “a pre-TC depression”. Below, we present some observational evidence and confirm that such impact exists in nature, and it is consistent with current dynamical understandings.

165 Genesis Potential Index (GPI) sets up a bridge between various environment variables and the *potential* for TC genesis over a large region (such as the northwestern Pacific in this study). Therefore, the projection of TC genesis variation under climate change is different from the prediction of an individual TC genesis potential.

170 GPI is usually applied with low-frequency data, such as monthly data. The reviewer is correct that SST is critical for the genesis of an individual TC. Nevertheless, for the potential of TC genesis, the regional mean SST is not the only oceanic contributor. Instead, a deeper D_{26} provides a favorable environment and a higher probability for TC genesis. Gray (1979) argued that “other conditions being favorable and remaining constant, seasonal tropical cyclone genesis frequency should be directly related to the magnitude of ocean thermal energy E ”, and “thermal potential may also be thought as the potential for Cb (Cumulonimbus) convection.” Hence, he used “ocean thermal energy” instead of SST as the key parameter for TC genesis, because the ocean provides
175 energy for TC genesis and the 2-D SST does not contain mass or energy (re-emphasized in Gray 1998, ref #24 in the manuscript). In a very recent paper, Emanuel (2022) confirmed again that “the annual mean frequency of tropical cyclones depends on climate parameters, such as the specified ocean mixed layer depth”. Therefore, for the potential genesis of TCs, the thermal energy in the
180 upper ocean (which is represented with D_{26}) can play a non-negligible role.

Moreover, in many tropical regions, SST is a good proxy for ocean thermal energy, since it has a good correlation with D_{26} as shown in Fig. 4B of the manuscript. Nevertheless, such an

assumption does not hold for the central-north Pacific (yellow box in Figs. 3 and 4A of the manuscript). The opposite relations between SST and D_{26} in the western north Pacific and the central-north Pacific was highlighted in this study (Fig. 4B and 4C). As a result, SST alone is no longer a good representation of the ocean thermal energy, and D_{26} is found to be a necessary supplement in this situation.

Practically, the dependence of TC genesis numbers on D_{26} can be seen from Fig. R3. For example, when \bar{T} is between 29°C and 30°C, the TC genesis numbers increase significantly when D_{26} deepens from 40 m to 120 m. The results remain qualitatively the same if the bin size of \bar{T} is reduced.

More light can be shed onto this issue by applying the Ekman theory and JTWC data. As shown in the Methods section of the manuscript, the Ekman pumping velocity (w_e) is calculated by

$$w_e = -\frac{1}{\rho} \left[\nabla \times \left(\frac{\vec{\tau}}{f} \right) \right] \dots \text{(R1)},$$

where $\rho = 1025 \text{ kg/m}^3$ is the mean seawater density; f is the Coriolis parameter; and $\vec{\tau}$ is the surface wind stress. $\vec{\tau}$ is estimated with the bulk formula $\rho_{air} C_D |\vec{U}| \vec{U}$, where $\rho_{air} = 1.2 \text{ kg/m}^3$ is the air density; C_D is the drag coefficient; and \vec{U} is the sea surface wind velocity. Assuming a circular wind structure of a tropical cyclone, $\nabla \times \vec{\tau}$ can be estimated with the maximum sustained wind speed (\vec{U}_{max}) and the radius of maximum wind (R). For simplicity, assuming an f -plan which only introduces limited quantitative errors, one can obtain

$$w_e = \frac{\rho_{air} C_D |\vec{U}_{max}|^2}{\rho f R} \dots \text{(R2)}.$$

The drag coefficient C_D has a large uncertainty at high wind speeds and it varies from basin to basin. Kara et al. (2007) concluded that $C_D \geq 1.5 \times 10^{-3}$ for the North Pacific. Here, we set C_D to the lower bound of 1.5×10^{-3} , so that w_e tends to be underestimated below. From 1980 to 2020, \vec{U}_{max} and R are available in JTWC for 422 tropical storms in the western North Pacific when they reached 35 knots for the first time. The mean w_e of the 422 cases is $2.9 \times 10^{-4} \text{ m/s}$, which indicates that water can rise from 80 m deep (the mean D_{26} used in this study) to the sea surface in merely 3 days.

210 Overall, both in the physical and statistical analyses, D_{26} tends to have a significant impact
 on TC genesis. It is possible that this impact may not be universal. Nevertheless, the contrast in
 TC genesis numbers between El Niño and La Niña over the northwestern Pacific is very evident,
 which is the motivation of this study. We also agree with the reviewer that the dynamic bonds
 between the ocean subsurface and TC genesis are far from obvious. Much more dedicated work is
 215 required in the future for a better understanding of the observations (Fig. R3).

In the revised manuscript, Fig. R3 is added as the new Fig. S1.

Figure R3 Joint distribution of TC genesis numbers with respect to \bar{T} and D_{26} . More TCs are born
 when D_{26} is deeper given the same \bar{T} . The conclusion stays valid even if the bin size of \bar{T} reduces.

220 **#4 There is a shift of pattern TC genesis in the central to western north Pacific as shown In
 Fig. 2. But this shift can be more likely due to shift in vertical wind-shear patterns rather
 than due to D_{26} . Surface wind patterns shown in Fig. 5A would indicate a change in vertical
 shear in the region about 5-15N and in the so-called green and yellow box which may have
 225 different contributions to G function variations. If the authors formulated the G function
 with considerations of vertical shear instead of D_{26} , the changes in G function will be
 attributed to something else other than D_{26} . Thus the inference draw from G functional
 analysis are nonunique.**

Reply:

230 We agree with the reviewer that the vertical wind shear is usually important for TC genesis,
and it has been proven to play a key role in modulating TC genesis during ENSO, especially in the
Atlantic Ocean (Camargo et al. 2007, i.e., ref # 19).

Although the vertical wind shear was not explicitly included in GPI_o (not predetermined but
selected by the regression algorithm; Fig. R1), it does not mean that the effect of vertical wind
235 shear was excluded. The reviewer is correct that the effect of vertical wind shear was represented
by other variables that were already included in GPI_o . In fact, SST is the highly related one, rather
than D_{26} . SST is shown to be a key parameter in controlling the vertical wind shear (Latif et al.
2007). As Fig. R4A shows, the pattern of the changes in the vertical wind shear from La Niña to
El Niño is close to that of \bar{T} (Fig. 3A in the manuscript). This is further supported by Fig. R4B.
240 There are significantly negative correlations between the vertical wind shear and \bar{T} , except for the
upper-left corner of the green box. Note that strong vertical wind shear is unfavorable for TC
genesis. Thus, the impact of vertical wind shear and \bar{T} on TC genesis is positively correlated. In
other words, the impact of vertical wind shear can be largely represented by the \bar{T} term in GPI.
That's probably why the vertical wind shear was not included in any of GPI_o and GPI_{ao} in Zhang
245 et al. (2016). Besides, the vertical wind shear is relatively less important for TC genesis in the
Pacific than in the Atlantic (Camargo et al. 2007, i.e., ref # 19; Fu et al. 2012; Peng et al. 2012).
As Fig. R5 shows, the climatological wind shear is less than 10 m/s over most northwestern Pacific
Ocean, which cannot prevent the TC genesis (Bracken and Bosart 2000). That is another reason
that the vertical wind shear was not selected into the GPI developed for the northwestern Pacific.

250 In summary, the effect of vertical wind shear was not really excluded in this study. It is
implicitly represented since its impact is closely related to that of \bar{T} , but there is no evidence that
the impact of D_{26} can be represented by the vertical wind shear.

The above explanations are briefly summarized on Lines 69-72 in the revised manuscript.

255 **Figure R4 (A)** The differences of vertical wind shear between El Niño and La Niña. **(B)** The correlation coefficients between the vertical wind shear anomalies and \bar{T} anomalies.

260 **Figure R5** The climatological vertical wind shear between 200 hPa and 850 hPa from July to October. The unit is m/s.

#5 The hypothesis about importance of D_{26} has to be further supported such as by CGCM simulations to demonstrate that D_{26} really matters.

Reply:

265 We agree that the conclusions in this study require further verifications by CGCM.

270 Firstly, we evaluated the simulated relations between D_{26} and \bar{T} in 20 CMIP6 models and 10 CMIP6 HighResMIP (Roberts et al. 2020) models. As shown in Fig. 4C of the manuscript, D_{26} and \bar{T} have a negative correlation in the central-north Pacific in observations. This is reproduced with a black bar in Figs. R6 and R7. In 20 CMIP6 models (Fig. R6), 12 models produce positive correlations, and 8 models capture the negative correlations. The lowest negative correlation coefficient is -0.22 from CESM2, which is still much weaker than the observation (~ -0.4). In 10 CMIP6 HighResMIP models (Fig. R7), only two models, i.e., EC-Earth3P-HR and ECMWF-IFS-HR, reproduce negative correlations albeit correlations close to zero. Therefore, it can be concluded that the relation between D_{26} and \bar{T} over the central-north Pacific is a common bias in model simulations.

280 Secondly, we examine the dependence of TC genesis on D_{26} using the products from 10 CMIP6 HighResMIP models. As Fig. R8 shows, a larger correlation coefficient between D_{26} and SST over the central-north Pacific leads to an increase of TC genesis number during El Niño compared with that during La Niña. In the 10 CMIP6 HighResMIP models, the simulated SSTs increase over the central-north Pacific during El Niño as expected. Thus, a shallow D_{26} (i.e., a

weak positive correlation between D_{26} and SST) can reduce the tendency of increase in TC genesis number during El Niño, which lend supports to the conclusions in this study.

In fact, at present, only a few global climate models can resolve and reproduce TCs (such as Zhao et al. 2009). It is common that a CGCM produces fewer TCs than in observations (Camargo 2013). Actually, the reason why GPIs are practically useful is partly due to the limited reproduction of TCs in contemporary climate models. Therefore, current conclusions are but the first critical step and serve as a call for more dedicated verifications with high-resolution CGCMs in the future.

The information about HighResMIP is added on Lines 151-152 in the revised manuscript.

290 **Figure R6** Correlation coefficient between D_{26} and \bar{T} in the central-north Pacific (yellow box in the manuscript) in observations (black column) and in 20 CMIP6 models in historical experiment (gray columns).

Figure R7 Same as Fig. R8 but for 10 CMIP6 HighResMIP models in the hist-1950 experiment.

295

Figure R8 Dependence of the increased TC genesis numbers during El Niño on the relation between D_{26} and SST over the central-north Pacific. The x-axis shows the correlation coefficient between D_{26} and SST over the central-north Pacific. The y-axis is the increase of TC genesis number between El Niño than La Niña in the whole western North Pacific (110°E – 170°W , 0° – 30°N), normalized by the TC genesis numbers during La Niña. The regression coefficient is significant at the 95% confidence level.

300

305 **#6 The authors tends to stress "The negative correlation between warm SST anomalies and shallower 26C isotherm depth is opposite to the current understanding for ENSO dynamics" . But this has little to do with ENSO dynamics. The difference in green and yellow box in terms of Tsub and SST is more related to the difference in heat flux due to wind evaporations, which can become clear if one adds the wind pattern in Fig. 5A onto the summer climatology wind, at least to my eyeballing estimations.**

310 Reply:

The relation between SST and thermocline depth is a key component in ENSO dynamics. For example, in the recharge-discharge theory (Jin 1997), SST in the eastern Pacific (T_E) was represented by (Eq. 2.4 in Jin 1997)

$$\frac{dT_E}{dt} = -cT_E + \gamma h_E + \delta_s \tau_E,$$

315 where h_E is the thermocline depth and τ_E is the surface wind stress. A deep h_E leads to a warm anomaly of T_E . In this study, the contrast between Figs. 4B and 4C in the manuscript indicated that the relation between D_{26} (similar to the thermocline depth) and SST was different from the one in the eastern Pacific. Note that the negative correlation between D_{26} and SST occurs over the central-north Pacific, which is beyond the two boxes prescribed in the classical recharge-discharge theory.

320 The relations between the simulated ENSO and the $D_{26}-\bar{T}$ correlation in the central-north Pacific are examined using 20 CMIP6 models. The simulated ENSO is evaluated with the CLIVAR 2020 ENSO Metrics Package (Planton et al. 2021). The package was developed by PCMDI, IPSL/LOCEAN, and NOAA. One metric is the τ_x -SSH feedback, i.e., the regression slope of sea surface height (SSH) anomalies (as a proxy for subsurface temperature) in the Niño3 region onto the zonal wind stress (τ_x) anomalies in the Niño4 region. The feedback is the ocean-atmospheric branch of the Bjerknes feedback as defined in Planton et al. (2021), connecting the trade wind anomalies and the subsurface temperature anomalies in the eastern equatorial Pacific. Such feedback is generally weaker in CMIP6 models than that in the observation. The ENSO simulations are evaluated by the score,

$$\text{score} = \text{abs} \left(\frac{m - r}{r} \right) \dots \text{(R3)},$$

where m is the regression coefficient in simulations; r is the regression coefficient in observations. The score actually measures the difference between simulations and observations, normalized by

the observations. A model has a better simulation of the feedback, if the score approaches zero.
 335 Therefore, this score is also a metric for ENSO simulations. The score for the simulated relation
 between D_{26} and \bar{T} over the central-north Pacific is similarly defined.

The relations between the $D_{26}-\bar{T}$ score and the ENSO simulations in CMIP6 models are
 shown in Fig. R9. The x-axis shows the score (Eq. R3) for the regression between D_{26} and \bar{T} over
 the central-north Pacific (yellow box in the manuscript) in the models. The y-axis shows the score
 340 in Eq. R3 for ENSO simulations. The significantly positive regression coefficient in Fig. R9
 indicates that a better simulation of the $D_{26}-\bar{T}$ relation over the central-north Pacific is
 accompanied with a better ENSO simulation in CMIP6 models.

Overall, as the reviewer states, the dynamic implication of the negative correlation between
 D_{26} and \bar{T} over the central-north Pacific to ENSO is not fully explored in this study. Further
 345 dynamical research is desired, but it is beyond the scope of current study.

Figure R9 Association between the $D_{26}-\bar{T}$ relation over the central-north Pacific and the ENSO
 simulation in 20 CMIP6 models. The x-axis shows the scores for simulated $D_{26}-\bar{T}$ relation. The
 y-axis shows the scores for ENSO simulations. Both scores are defined in Eq. R3. The correlation
 350 coefficient is significant at the 99% confidence level.

I do not rule out completely the role of subsurface ocean in GPI. But I provide my assessments to convince the authors that they need to provide direct evidence to prove (or falsify) the hypothesis that the D_{26} matters for GPI. The empirical G_s are not adequate and easily misleading as I can see into it. As I personally know the authors well, I am signing this negative but otherwise constructive review (with an interesting mathematical analysis which potentially can be a useful tool itself for the authors to use).

Reply:

We are happy that the reviewer agrees that the subsurface ocean can play a role in TC genesis. In the above responses and the revised manuscript, we added much more direct evidence to verify the hypothesis that the D_{26} matters for TC genesis. We would like to emphasize that most evidence does not depend on the GPI. Instead, they are obtained from observations, such as Figs. 1, 4, 5, S1, S3, S5 in the revised manuscript. The GPI is only a quantitative tool, whose results are consistent with the observations. Therefore, we believe our conclusions capture the intrinsic properties of TC genesis in nature.

We truly appreciate the reviewer for his time, honest evaluation, mathematical analysis, and the very insightful comments. We hope that he finds our responses and revisions acceptable and complete.

370 **Reviewer #2 (Remarks to the Author):**

In this study, the authors highlight a counterintuitive relationship between the potential TC genesis in the Central North Pacific and the phase of ENSO. Whereas the warm El Niño phase is characterized by an extension of the warm pool and warmer surface anomalies in the Central-north Pacific, the depth of the isoT26C is actually shallower (due to increased Ekman suction in relation with the large scale wind stress anomalies associated with the North Pacific subtropical high) making the heat content lower and counteracting the favorable sea surface environment for TC activity which leads to a suppression of TC genesis in this region. While the idea and proposed mechanism are interesting, I think the conclusion is a bit overstated. The authors claimed a suppression, whereas it is merely a reduction of the TC genesis potential. This study is well written and has the potential for publication in Nature Communication but would require more convincing analysis or a shift in focus beforehand.

385 **Major comments:**

1/ Figure 2C displays actually an increase in TC genesis during El Niño phases compared to La Niña's. In addition, I wonder if the unchanged TC number between the 2 phases of ENSO shown in their figure 1 comes from the choice of their averaging region (not clearly specified by the way, but which, I assume, encompasses the whole warm pool) that integrates the reduction of TC number around the Philippines and the increase between 140 and 170W and lead to an overall similar number of TC during both ENSO phases.

What if this figure was done using the region delineated by their yellow dashed box, would the increase in TC genesis during El Niño remain insignificant?

Reply:

395 The reviewer is correct that the influence of D_{26} on the TC genesis numbers during ENSO applies over the whole warm pool region (110°E–170°W, 0°–30°N) in the northern Pacific. For the western region (110°E–160°E, 0°–30°N), TC genesis number slightly decreases during El Niño (Fig. R10A), and it is negatively but insignificantly correlated with ONI. For the yellow dashed box (160°E–170°W, 5°N–20°N) in the manuscript, the TC genesis number increases moderately
400 (Fig. R10B), which is mainly attributable to the eastward extension of warm pool during El Niño.

This result is consistent with many previous studies, such as Wang and Chan (2002). Combining the two regions, the TC genesis numbers show no significant differences between El Niño and La Niña over the tropical northwestern Pacific, as shown in Fig. 1B in the manuscript.

405 **Figure R10** Same as Fig. 1B in the manuscript but for TC genesis (A) to the west of 160°E and (B) in the central-north Pacific (yellow box in the manuscript).

2/ Maybe the authors could explore more the limitation in increase/reduction of potential TC genesis during El Niño/La Niña phases compared to climatological/neutral ENSO conditions?

410 For instance, the figure 2 of Lin et al. (2020) shows a limited increase/reduction in TC activity during El Niño/La Niña in the Central North Pacific compared to the westernmost part of the basin that might come from the proposed competing mechanism between SST and heat content:

- A limitation in TC genesis in that region during El Niño despite positive SST anomalies due to the shoaling of D_{26}
- the opposite during La Niña due to a reduction of SST anomalies but an increase in heat content

Reply:

420 We thank the reviewer for the insightful suggestions. Figure 17.2 in Lin et al. (2020) contains TC genesis and track information. Figure 17.3a shows the annual number of TCs in the WNP from 1960 to 2016. We reproduced Fig. 17.3a and present it as Fig. R11 below. Lin et al. (2020) state that “The increase in TC frequency during El Niño years ...”. It is true since the mean annual TC number of the 12 El Niño years is 26.42, while the mean annual TC number of the 15 La Niña

years is 25.13. However, the p -value is 0.5334, which indicates that the TC genesis increase during El Niño is not statistically significant. It is also true that “TC frequency during CP El Niño years is also above normal”. The mean annual TC number of 6 CP El Niño years is 29.33, and the mean annual TC number of 57 years (1960–2016) is 26.96. But the increase is still not significant, since the p -value is 0.3798. Therefore, current conclusion based on Fig. 1B in the manuscript is consistent with the results in Lin et al. (2020).

430 Following the reviewer’s suggestion, the differences in GPIs between El Niño and the climatology are shown in Fig. R12A; the differences between La Niña and the climatology are shown in Fig. R12B. Such differences are almost half of those between El Niño and La Niña, which are shown in Fig. 2C in the manuscript.

435 Therefore, comparing with the westernmost part of the basin, the two contents proposed by the reviewer are verified. The results are also consistent with the conclusions presented in the manuscript.

Figure R11 Annual TC numbers in the WNP (1960–2016). The figure is a reproduction of Fig. 17.3 in Lin et al. (2020).

440

Figure R12 (A) The differences in GPI between El Niño and the climatology. **(B)** The differences in GPI between La Nina and the climatology.

445

3/ Also, why not extending the analysis to the entire central Pacific (i.e. until 150°W, the usual definition of the Central North Pacific TC basin)?

Reply:

There are two reasons for the domain selection.

450

First, GPI is a statistical constraint between various environmental variables and the TC genesis number. The statistical relation varies with regions. Particularly, GPI_o was created for the northwestern Pacific Ocean (Zhang et al. 2016) and it performs well in this region. Such regional dependence of GPI is systematically described in Raavi and Walsh (2020). For example, vertical wind shear is important for TC genesis in the North Atlantic, but not as important as other parameters in the western North Pacific (Fu et al. 2012; Peng et al. 2012). Overall, GPI_o is an appropriate GPI for the northwestern Pacific, but might not work for the eastern tropical Pacific.

455

Second, as shown in Fig. R13, most TCs that are generated to the west of 150°W travel to the west, leaving a trail of devastation over eastern Asia. While most TCs generated to the east of 150°W travel to the east and have impacts on the Americas. Hence, practically, it is meaningful and useful to explore the slow variation of TC genesis separately between the northwestern and the eastern Pacific Ocean. Figure 1B in the manuscript can be reproduced to the west of 150°W, following the reviewer’s suggestion. The results are consistent with the original figure, since TC genesis numbers between 170°W and 150°W are small (Fig. R13).

460

More information is added to item *a* in the Methods section.

Figure R13 Tropical cyclone genesis positions from 1980 to 2022. The red dots represent the TCs which moved toward the eastern Asia; the blue dots represent the TCs travelling to the east.

4/ Knowing that the heat content is more influential on TC intensification than genesis, I wonder if these results would be changed (even maybe strengthened) if the authors used an index accounting for TC intensity (the Accumulated Cyclone Energy for instance, Bell et al 2000, 2004) as compared to an index of TC genesis or limited their analysis to the occurrence of major TC (Category 3 and above) in this region?

Reply:

We appreciate the reviewer’s very helpful and feasible suggestion. We fully agree that “the heat content is more influential on TC intensification than genesis”. It will be very interesting to quantitatively explore such an impact on TC intensity.

Following the reviewer’s suggestion, we computed the accumulated cyclone energy (ACE), which is shown in Fig. R14. There is a significantly positive correlation between ONI and ACE. ACE depends both on TC frequency and TC intensity as well as TC duration. During El Niño, more TCs are born over the central Pacific, although the total TC genesis numbers do not increase (based on current results in the manuscript). Therefore, the TCs tend to have more time to intensify over the ocean before landing over eastern Asia. As a result, more cyclone energy can be accumulated. Hence, the increase of ACE during El Niño shown in Fig. R14 has no conflict with the conclusions in the manuscript. Moreover, we also counted TCs in the categories from 1 to 5

(Fig. R15). More intensive TCs occur during El Niño than during La Niña, which is consistent
485 with the changes in ACE (Fig. R14) and the findings in Camargo and Sobel (2005).

Existing studies have confirmed that the subsurface ocean environments have important
influences on TC intensity. Lin et al. (2013) modified the potential intensity index by replacing
SST with depth-averaged ocean temperature to quantify the role of subsurface temperature on TC's
intensity. Zheng et al. (2015) unveiled the unfavorable oceanic impacts during El Niño. Particularly,
490 according to Huang et al. (2015). It is likely that D_{26} can have a non-negligible impact on TC
intensity. However, in order to quantify such an impact, an index for TC intensity needs to be
created following the algorithm shown in Fig. R1 (see Zhang et al. 2016 for more details). In this
way, all variables which may be influential on TC intensity will be considered, and an index for
TC intensity is created with the recursive regression method. Based on our current dynamical
495 understanding (Huang et al. 2015), it is highly probable that D_{26} would be selected as a variable
in the new index. Then, the impact of D_{26} on TC intensity in an ocean basin can be quantitatively
estimated. One can see that this is a separate though related study from the current one. We will
complete that in the future and report the results separately.

This issue is addressed at the end of the manuscript by saying “our results do not exclude the
500 possibility that super typhoons may increase in number, since more TCs tend to be generated over
the tropical central Pacific and can have a longer lifetime to grow over the warm ocean”.

Figure R14 The same as Fig. 1B in the manuscript but for the accumulated cyclone energy (ACE).
The unit is $10^5 \text{ m}^2/\text{s}^2$. The correlation coefficient is significant at the 99% confidence level.

505

Figure R15 The monthly mean TC numbers of Category 1 to 5 during La Niña, Neutral, and El Niño conditions.

510 **Minor comments:**

Reference 13 is not the best suited

Reply:

Ref #13 is replaced with Chapter “Impact of El Niño on Weather and Climate Extremes” from the book “El Niño Southern Oscillation in a Changing Climate” (Goddard and Gershunov 2020).

515

Line 42: I thought the term had been coined “hiatus”

Reply:

“global warming pause” has been replaced with “hiatus”.

520 **Line 53: Phase not flavor**

Reply:

Here we intended to say different types of ENSO, such as the CP El Niño and the EP El Niño. In the revised manuscript, “flavor” is replaced with “type”.

525

References

- Bracken, W. E., and L. F. Bosart, 2000: The Role of Synoptic-Scale Flow during Tropical Cyclogenesis over the North Atlantic Ocean. *Mon. Wea. Rev.*, **128**, 353-376.
- 530 Camargo, S. J., and A. H. Sobel, 2005: Western North Pacific tropical cyclone intensity and ENSO. *J. Clim.*, **18**, 2996-3006.
- Camargo, S. J., K. A. Emanuel, and A. H. Sobel, 2007: Use of a Genesis Potential Index to Diagnose ENSO Effects on Tropical Cyclone Genesis. *J. Clim.*, **20**, 4819-4834.
- Camargo, S. J., 2013: Global and Regional Aspects of Tropical Cyclone Activity in the CMIP5 Models. *J. Clim.*, **26**, 9880-9902.
- 535 Cao, J., H. Zhao, B. Wang, and L. Wu, 2021: Hemisphere-asymmetric tropical cyclones response to anthropogenic aerosol forcing. *Nature Communications*, **12**, 6787.
- DeMaria, M., and J. Kaplan, 1994: A Statistical Hurricane Intensity Prediction Scheme (SHIPS) for the Atlantic Basin. *Weather and Forecasting*, **9**, 209-220.
- 540 DeMaria, M., M. Mainelli, L. K. Shay, J. A. Knaff, and J. Kaplan, 2005: Further Improvements to the Statistical Hurricane Intensity Prediction Scheme (SHIPS). *Weather and Forecasting*, **20**, 531-543.
- Emanuel, K., 2022: Tropical Cyclone Seeds, Transition Probabilities, and Genesis. *J. Clim.*, **35**, 3557-3566.
- 545 Emanuel, K. A., and D. Nolan, 2004: Tropical cyclone activity and the global climate system. *Preprints, 26th Conf. on Hurricanes and Tropical Meteorology, Miami, FL, Amer. Meteor. Soc. A.*
- Emanuel, K. A., 2013: Downscaling CMIP5 climate models shows increased tropical cyclone activity over the 21st century. *Proc. Natl. Acad. Sci.*, **110**, 12219-12224.
- 550 Ford, W., 2015: Chapter 14 - Gram-Schmidt Orthonormalization. *Numerical Linear Algebra with Applications*, W. Ford, Ed., Academic Press, 281-297.
- Fu, B., M. S. Peng, T. Li, and D. E. Stevens, 2012: Developing versus Nondeveloping Disturbances for Tropical Cyclone Formation. Part II: Western North Pacific. *Mon. Wea. Rev.*, **140**, 1067-1080.
- 555 Fu, D., P. Chang, C. M. Patricola, R. Saravanan, X. Liu, and H. E. Beck, 2021: Central American mountains inhibit eastern North Pacific seasonal tropical cyclone activity. *Nature Communications*, **12**, 4422.
- Goddard, L., and A. Gershunov, 2020: Impact of El Niño on Weather and Climate Extremes. *El Niño Southern Oscillation in a Changing Climate*, 361-375.
- 560 Huang, P., I. I. Lin, C. Chou, and R.-H. Huang, 2015: Change in ocean subsurface environment to suppress tropical cyclone intensification under global warming. *Nat. Commun.*, **6**, 7188.
- Jin, F. F., 1997: An equatorial ocean recharge paradigm for ENSO .1. Conceptual model. *J. Atmos. Sci.*, **54**, 811-829.
- 565 Kara, A. B., A. J. Wallcraft, E. J. Metzger, H. E. Hurlburt, and C. W. Fairall, 2007: Wind Stress Drag Coefficient over the Global Ocean. *Journal of Climate*, **20**, 5856-5864.
- Knaff, J. A., C. R. Sampson, and M. DeMaria, 2005: An Operational Statistical Typhoon Intensity Prediction Scheme for the Western North Pacific. *Weather and Forecasting*, **20**, 688-699.
- Latif, M., N. Keenlyside, and J. Bader, 2007: Tropical sea surface temperature, vertical wind shear, and hurricane development. *Geophys. Res. Lett.*, **34**.
- 570 Lavender, S. L., and Coauthors, 2018: Estimation of the maximum annual number of North Atlantic tropical cyclones using climate models. *Science Advances*, **4**, eaat6509.

Lee, C.-Y., S. J. Camargo, A. H. Sobel, and M. K. Tippett, 2020: Statistical–Dynamical Downscaling Projections of Tropical Cyclone Activity in a Warming Climate: Two Diverging Genesis Scenarios. *J. Clim.*, **33**, 4815–4834.

575 Lin, I.-I., and Coauthors, 2013: An ocean coupling potential intensity index for tropical cyclones. *Geophys. Res. Lett.*, **40**, 1878–1882.

Murakami, H., and B. Wang, 2010: Future change of North Atlantic tropical cyclone tracks: Projection by a 20-km-mesh global atmospheric model. *J. Clim.*, **23**, 2699–2721.

580 Murakami, H., 2022: Substantial global influence of anthropogenic aerosols on tropical cyclones over the past 40 years. *Science Advances*, **8**, eabn9493.

Patricola, C. M., P. Chang, and R. Saravanan, 2016: Degree of simulated suppression of Atlantic tropical cyclones modulated by flavour of El Niño. *Nature Geoscience*, **9**, 155–160.

Peng, M. S., B. Fu, T. Li, and D. E. Stevens, 2012: Developing versus Nondeveloping Disturbances for Tropical Cyclone Formation. Part I: North Atlantic. *Mon. Wea. Rev.*, **140**, 1047–1066.

585 Planton, Y. Y., and Coauthors, 2021: Evaluating Climate Models with the CLIVAR 2020 ENSO Metrics Package. *Bull. Amer. Meteor. Soc.*, **102**, E193–E217.

Raavi, P. H., and K. J. E. Walsh, 2020: Basinwise Statistical Analysis of Factors Limiting Tropical Storm Formation From an Initial Tropical Circulation. *Journal of Geophysical Research: Atmospheres*, **125**, e2019JD032006.

590 Roberts, M. J., and Coauthors, 2020: Projected Future Changes in Tropical Cyclones Using the CMIP6 HighResMIP Multimodel Ensemble. *Geophys. Res. Lett.*, **47**, e2020GL088662.

Wang, B., and J. C. L. Chan, 2002: How Strong ENSO Events Affect Tropical Storm Activity over the Western North Pacific. *J. Clim.*, **15**, 1643–1658.

595 Yang, W., T.-L. Hsieh, and G. A. Vecchi, 2021: Hurricane annual cycle controlled by both seeds and genesis probability. *Proc. Natl. Acad. Sci.*, **118**, e2108397118.

Zhang, M., L. Zhou, D. Chen, and C. Wang, 2016: A genesis potential index for Western North Pacific tropical cyclones by using oceanic parameters. *Journal of Geophysical Research: Oceans*, **121**, 7176–7191.

600 Zhao, J., R. Zhan, Y. Wang, S.-P. Xie, and Q. Wu, 2020: Untangling impacts of global warming and Interdecadal Pacific Oscillation on long-term variability of North Pacific tropical cyclone track density. *Science Advances*, **6**, eaba6813.

Zhao, M., I. M. Held, S.-J. Lin, and G. A. Vecchi, 2009: Simulations of Global Hurricane Climatology, Interannual Variability, and Response to Global Warming Using a 50-km Resolution GCM. *J. Clim.*, **22**, 6653–6678.

605 Zheng, Z.-W., I.-I. Lin, B. Wang, H.-C. Huang, and C.-H. Chen, 2015: A long neglected damper in the El Niño—typhoon relationship: a ‘Gaia-like’ process. *Sci. Rep.*, **5**, 11103.

REVIEWER COMMENTS

Reviewer #1

The authors addressed most of my concerns. I am happy to recommend for its publication.

I do have two more suggestions.

As I am especial happy to see they indeed dogged into some existing high resolutions simulations to provide some modeling evidence about the potential effect of D26 on TC genesis (Fig. R8), I wish they do so more directly. This figure is suggestive but can be not directly compared with the observed results in the key figure of this paper (Fig3D). They can use the model simulations and their choice of GPI formula to produce a similar figure (multi-model composite or etc) as Fig.3D. This kind of figures provide more direct evidence and informative addition if the results are as truly clearly supportive.

I also wish they do include another figure like Fig3D but used GPI_{0a} formulation as well. The difference in exponents of the two formulations yield a definite difference in terms of the level of D26 effect as I demonstrated mathematically. Whether it is a factor 2 or 1.5 (or somewhere in between) difference may depend on how one linearizing the GPI. But this point of mine is beyond question as it clearly shown in Fig.R2. I urge the authors to include this information in the extend figures so that readers will be aware with this caveat, which leaves door open for someone to go further to find ways to eliminate this uncertainty.

Reviewer #2

Review 2 of

Suppression of Tropical Cyclone Genesis by Subsurface Environment in the Tropical Central-North Pacific during El Niño

by

Authors: Cong Gao¹, Lei Zhou^{1,2*}, Chunzai Wang³, I.-I. Lin⁴, Raghu Murtugudde^{5,6}

I appreciate the amount of work done by the authors during this round of review. My only concern now is that this work does not necessarily reflect in the new version of their manuscript.

I feel that results from Figures R10B and R14 represents a clear limitation to this study.

The authors claim a suppression of TC genesis in the Tropical central North Pacific (yellow box) based on their Figure 1B, but the region used for their calculation encompasses the whole North West Pacific TC basin (i.e. the Warm Pool). Meanwhile, Figure R10B shows a significant increase in TC genesis in the yellow box actually representing the Tropical Central-North Pacific. Therefore, I feel that “suppression” is an overstatement and that the title of the paper should be changed to something less misleading like:

Unexpected limitation of Tropical Cyclone Genesis by Subsurface Environment in the Tropical Central-North Pacific during El Niño

Regardless, I think the language of the manuscript still needs to be toned down to promote a more delicate message being that, based on the current understanding of ENSO dynamics, one would expect a much larger TC genesis in the Central-North Pacific during El Niño because of the Warm Pool’s extension, but this is actually not the case due to the anti-correlated behavior between SST and heat content in this region. This in itself is a rather important message in particular in the context of global warming as stated by the authors.

Also, I would like to see an extended discussion or a new “limitation section” on the fact that the limitation of TC genesis during El Niño in the tropical central-north Pacific does not reflect for intense TC (Cat. 3 and above). The sentence added at the end of the manuscript is too short and vague. On that matter, I would personally also appreciate seeing the Figure R14 calculated over the yellow box only.

As for the relevance of the math behind the use of GPI_{ocean} to assess the dominant role of D26 in controlling the TC genesis in the tropical central north Pacific, I leave it to Prof. Jin’s expert appreciation.

Responses to Reviewer 1

[General Comment] The authors addressed most of my concerns. I am happy to recommend for its publication.

We appreciate the reviewer for all the constructive suggestions and the positive comment.

5

[Comment 1] As I am especial happy to see they indeed dogged into some existing high resolutions simulations to provide some modeling evidence about the potential effect of D_{26} on TC genesis (Fig. R8), I wish they do so more directly. This figure is suggestive but can be not directly compared with the observed results in the key figure of this paper (Fig. 3D). They can use the model simulations and their choice of GPI formula to produce a similar figure (multi-model composite or etc) as Fig. 3D. This kind of figures provide more direct evidence and informative addition if the results are as truly clearly supportive.

Response:

Following the reviewer's suggestion, Fig. A1 is produced using the ten HighResMIP climate models. In agreement with Fig. 3D in the main text which is based on observations, Fig. A1 shows a nontrivial and negative impact of D_{26} on TC genesis numbers, including the central-north Pacific (the yellow box). Nevertheless, the negative impact of D_{26} in the yellow box is confined to the lower panel, instead of the whole box as in Fig. 3D. The discrepancy is likely to be largely attributable to the inadequate performance of the HighResMIP models in reproducing the relationship between D_{26} and \bar{T} as shown in Fig. R7 in the previous response. Such model deficiencies are not surprising and yet important to report.

15
20

Figure A1 is included as the Supplementary Fig. S7A in the revised manuscript.

Figure A1 Same as Fig. 3D in the main text, but obtained with the multi-model composite of 10 HighResMIP climate models. GPI_{ocean} is used for this figure.

25

[**Comment 2**] I also wish they do include another figure like Fig. 3D but used GPI_{oa} formulation as well. The difference in exponents of the two formulations yield a definite difference in terms of the level of D26 effect as I demonstrated mathematically. Whether it is a factor 2 or 1.5 (or
30 somewhere in between) difference may depend on how one linearizing the GPI. But this point of mine is beyond question as it clearly shown in Fig. R2. I urge the authors to include this information in the extend figures so that readers will be aware with this caveat, which leaves door open for someone to go further to find ways to eliminate this uncertainty.

Response:

35 Figure R2 in the previous response is added in the revised manuscript as Supplementary Fig. S8. Moreover, we also reproduce Fig. A1, which is obtained with the HighResMIP model outputs, using GPI_{oa}. The results are shown in Fig. A2, which is included as Supplementary Fig. S7B in the revised manuscript. We hope that the quantitative uncertainties due to different GPIs are clearly seen with the new Supplementary Figs. S7 and S8.

Figure A2 Same as Fig. A1, but calculated with GPI_{oa}.

Responses to Reviewer 2

45 [General Comment] I appreciate the amount of work done by the authors during this round of review. My only concern now is that this work does not necessarily reflect in the new version of their manuscript.

We thank the reviewer for appreciating our additional efforts. We are also thankful for all the constructive suggestions. We incorporate the new results and tone down the claims in the revised version. We hope the messages are expressed in an appropriate way after the revision and the
50 reviewer is satisfied.

[Comment 1] I feel that results from Figures R10B and R14 represents a clear limitation to this study. The authors claim a suppression of TC genesis in the Tropical central North Pacific (yellow box) based on their Figure 1B, but the region used for their calculation encompasses the whole
55 North West Pacific TC basin (i.e. the Warm Pool). Meanwhile, Figure R10B shows a significant increase in TC genesis in the yellow box actually representing the Tropical Central North Pacific. Therefore, I feel that “suppression” is an overstatement and that the title of the paper should be changed to something less misleading like: **Unexpected limitation of Tropical Cyclone Genesis by Subsurface Environment in the Tropical Central-North Pacific during El Niño.**

60 **Response:**

We agree with the reviewer that our current results have two limitations. One is that the results apply to the whole North West Pacific, rather than to the central-north Pacific as shown in Fig. R10B in the previous response. The other one is that the results apply to the genesis numbers, but not to other properties of TCs (such as ACE shown in Fig. R14 in the previous response). The two
65 limitations are mentioned in the Discussion section in the revised manuscript. Figure R15 in previous response is included in the revised manuscript as Supplementary Fig. S9A.

We are grateful to the reviewer for suggesting a more appropriate title. We adopt it in the revised version. Thank you.

70 [Comment 2] Regardless, I think the language of the manuscript still needs to be toned down to promote a more delicate message being that, based on the current understanding of ENSO

dynamics, one would expect a much larger TC genesis in the Central-North Pacific during El Niño because of the Warm Pool's extension, but this is actually not the case due to the anti-correlated behavior between SST and heat content in this region. This in itself is a rather important message
75 in particular in the context of global warming as stated by the authors.

Response:

We thank the reviewer for the succinct and accurate summary of our findings. That's exactly what our message is even though we may have overstated some aspects. We also thank the reviewer for the acknowledgement of the significance of our work. Following the reviewer's suggestion,
80 we revise the entire manuscript to ensure the results are not overstated.

[Comment 3] Also, I would like to see an extended discussion or a new "limitation section" on the fact that the limitation of TC genesis during El Niño in the tropical central-north Pacific does not reflect for intense TC (Cat. 3 and above). The sentence added at the end of the manuscript is
85 too short and vague. On that matter, I would personally also appreciate seeing the Figure R14 calculated over the yellow box only.

Response:

In the revised manuscript, more discussion on the limitations is added in the Discussion section, particularly on the two limitations mentioned above.

90 Following the reviewer's suggestion, the correlation between ACE and ONI only over the yellow box is shown in Fig. A3. The correlation coefficient and the fitted slope are smaller than the ones in Fig. R14 in previous response. In the revised manuscript, Fig. R14 in previous response and Fig. A3 below are merged as Supplementary Fig. S9.

95 **Figure A3** Same as Fig. R14 in previous response, but for the region within 5°N–20°N and 160°E–170°W (the yellow box) only. The unit of ACE is $10^5 \text{ m}^2 \text{ s}^{-2}$.

[Comment 4] As for the relevance of the math behind the use of GPIocean to assess the dominant role of D26 in controlling the TC genesis in the tropical central north Pacific, I leave it to Prof.

100 Jin’s expert appreciation.

We thank the reviewer for the specific and insightful suggestions again.

REVIEWERS' COMMENTS

Reviewer #1 (Remarks to the Author):

The authors have adequately addressed the my comments, I recommend its publication as it is.

Review 3 of

Unexpected limitation of Tropical Cyclone Genesis by Subsurface Environment in the Tropical Central-North Pacific during El Niño

by

Authors: Cong Gao¹, Lei Zhou^{1,2*}, Chunzai Wang³, I.-I. Lin⁴, Raghu Murtugudde^{5,6}

I thank the authors for recognizing the limitations of their study and for addressing them in the revised version. I now recommend publication of this article in Nature Communication, but with some proofreading to improve the wording and language. I suggest below a few rewording:

Minor comments:

Line 66: while -> whereas

Line 83-86: I recommend rephrasing to something like :” However, it is generally weaker than 10 m s-1 over most of the northwestern Pacific Ocean and thus generally not large enough to prohibit TC genesis. Indeed, vertical wind shear was tested but the GPlocean was not found to be as sensitive to it as the factors listed above and thus was not retained in the GPlocean calculation”

Line 86: over other GPs **formulation**

Line 98: with -> using

Line 99: is-> The changes represents...

Line 137: **described** above

Line 153: in **the** opposite direction